# Hydrogen bond unlocking-driven pore structure control for shifting multi-component gas separation function

Rong Yang[1,6], Yu Wang [1,6], Jian-Wei Cao [1], Zi-Ming Ye[2], Tony Pham [3], Katherine A. Forrest[3], Rajamani Krishna [4], Hongwei Chen[5], Libo Li [5], Bo-Kai Ling[1], Tao Zhang [1], Tong Gao[1], Xue Jiang[1], Xiang-Ou Xu[1], Qian-Hao Ye[1] & Kai-Jie Chen [1] ✉

Purification of ethylene ($C_2H_4$) as the most extensive and output chemical, from complex multi-components is of great significance but highly challenging. Herein we demonstrate that precise pore structure tuning by controlling the network hydrogen bonds in two highly-related porous coordination networks can shift the efficient $C_2H_4$ separation function from $C_2H_2/C_2H_4/C_2H_6$ ternary mixture to $CO_2/C_2H_2/C_2H_4/C_2H_6$ quaternary mixture system. Single-crystal X-ray diffraction revealed that the different amino groups on the triazolate ligands resulted in the change of the hydrogen bonding in the host network, which led to changes in the pore shape and pore chemistry. Gas adsorption isotherms, adsorption kinetics and gas-loaded crystal structure analysis indicated that the coordination network Zn-fa-atz (2) weakened the affinity for three C2 hydrocarbons synchronously including $C_2H_4$ but enhanced the $CO_2$ adsorption due to the optimized $CO_2$-host interaction and the faster $CO_2$ diffusion, leading to effective $C_2H_4$ production from the $CO_2/C_2H_2/C_2H_4/C_2H_6$ mixture in one step based on the experimental and simulated breakthrough data. Moreover, it can be shaped into spherical pellets with maintained porosity and separation performance.

Separation of valuable components for multicomponent mixtures in one step without preconcentration is one of the most challenging tasks in separation science. In the process of $C_2H_4$ production, the product compositions of the thermal decomposition are complicated, and the conversion of dehydrogenation is only *ca.* 50%–60%[1]. Therefore, the resulting $C_2H_4$ usually contains a variety of impurities, among which carbon dioxide ($CO_2$), acetylene ($C_2H_2$) and ethane ($C_2H_6$) are the most difficult ones to separate because of very similar physical and chemical properties with $C_2H_4$[2,3]. To obtain polymer-grade $C_2H_4$ (>99.9%), multi-step processes are needed to remove the impurities, including chemical absorption, catalytic hydrogenation, cryogenic distillation, etc[4]. The stepwise purification processes result in huge equipment costs and energy consumption[5].

Using recyclable physisorbents, the adsorptive separation can be a promising approach for high-purity $C_2H_4$ thanks to the simple operation processes and lower energy requirements[6–10]. Metal–organic

[1]Key Laboratory of Special Functional and Smart Polymer Materials of Ministry of Industry and Information Technology, Xi'an Key Laboratory of Functional Organic Porous Materials, School of Chemistry and Chemical Engineering, Northwestern Polytechnical University, Xi'an, Shaanxi 710072, PR China. [2]Fujian Key Laboratory of Polymer Materials, College of Chemistry and Materials Science, Fujian Normal University, Fuzhou 350007, PR China. [3]Department of Chemistry, University of South Florida, Tampa, FL, USA. [4]Van 't Hoff Institute for Molecular Sciences, University of Amsterdam, Science Park 904, 1098 XH Amsterdam, The Netherlands. [5]Shanxi Key Laboratory of Gas Energy Efficient and Clean Utilization, College of Chemical Engineering and Technology, Taiyuan University of Technology, Taiyuan 030024, PR China. [6]These authors contributed equally: Rong Yang, Yu Wang. ✉e-mail: ckjiscon@nwpu.edu.cn

frameworks (MOFs), or porous coordination polymers (PCPs)/metal–organic materials (MOMs), with tunable pore structures[11–15], have shown great potential for binary C2 hydrocarbons separation, such as $C_2H_2$/$C_2H_4$[16–23], $C_2H_4$/$C_2H_6$[24–31], and $C_2H_2$/$CO_2$[32–36]. Compared with the multi-step separation process, purification of $C_2H_4$ in one-step from complex systems is more valuable in terms of energy utilization and chemical process. However, limited by the physicochemical properties of four gas molecules (kinetic diameter: $CO_2 \approx C_2H_2 < C_2H_4 < C_2H_6$; quadruple moment: $C_2H_2 > CO_2 > C_2H_4 > C_2H_6$)[37–39], it is extremely difficult to separate $C_2H_4$ from the quaternary $CO_2$/$C_2H_2$/$C_2H_4$/$C_2H_6$ in one step. Although a few of studies have achieved the one-step preparation of $C_2H_4$ from the ternary $C_2H_2$/$C_2H_4$/$C_2H_6$[40–48] or more difficult four-component separation[2,49,50], the understanding of such a complex systems and the corresponding principle of structural design are far from sufficiency[51].

Herein, we show that unlocking the framework hydrogen bonding can affect the pore size/shape and pore chemistry, and weaken the affinity to C2 hydrocarbons, especially $C_2H_4$ (Fig. 1). The fine turning of pore structure shifts the multi-component gas separation function, enabling one-step production of high-purity $C_2H_4$ in the quaternary $CO_2$/$C_2H_2$/$C_2H_4$/$C_2H_6$.

## Results

### Structure and adsorption properties of Zn-fa-datz (1)

[Zn$_2$(fa)(datz)$_2$] (Zn-fa-datz (1), H$_2$fa = fumaric acid, Hdatz = 1$H$-1,2,4-triazole-3,5-diamino) was initially selected[52], because of its high stability in moisture conditions (Supplementary Fig. 5), ultramicroporous nature and polar pore surface without open-metal coordination sites, based on our previously raised general rule[51]. Zn-fa-datz (1) is a pillared-layer coordination network with **pcu** topology (Supplementary Fig. 1). Each $Zn^{2+}$ ion is saturated by three N atoms from three datz$^-$ ligands and one O atom from a fa$^{2-}$ ligand, forming a 3D pillar-layered network with accessible 1D ultramicroporous channels (Fig. 2a and Supplementary Fig. 2). The purity and porosity were confirmed by powder X-ray diffraction (PXRD) pattern and 195 K $CO_2$ adsorption isotherm, respectively (Fig. 3a, Supplementary Fig. 3 and Supplementary Table 1). Note that, because $N_2$ diffuses extremely slowly in Zn-fa-datz (1) (Supplementary Fig. 4), 195 K $CO_2$ adsorption isotherm was conducted for the study of the porosity. As we expected, Zn-fa-datz (1) features stronger affinity for $C_2H_2$ (34.7 kJ mol$^{-1}$) and $C_2H_6$ (39.4 kJ mol$^{-1}$) than $C_2H_4$ (33.6 kJ mol$^{-1}$) at the low loading (Fig. 3d, Supplementary Figs. 6–9 and Supplementary Table 2). The equimolar $C_2H_2$/$C_2H_4$/$C_2H_6$ mixture breakthrough experiment shows that $C_2H_4$

eluted preferentially with high purity (99.9%) from three gases in the fixed-bed adsorber (Fig. 5a), thus further demonstrating that Zn-fa-datz (1) can achieve one-step purification of $C_2H_4$ in the ternary C2 hydrocarbon mixture. However, due to the lower $CO_2$ affinity (24.0 kJ mol$^{-1}$) than for $C_2H_4$ (33.6 kJ mol$^{-1}$) (Fig. 3c, d and Supplementary Table 2), Zn-fa-datz (1) failed to produce $C_2H_4$ in one-step from the equimolar $CO_2$/$C_2H_2$/$C_2H_4$/$C_2H_6$ quaternary mixture (Fig. 5b).

Regarding the thermodynamic aspect, the adsorption affinity for $C_2H_4$ should be the lowest among the four adsorbates in order to achieve one-step purification of $C_2H_4$ from the $CO_2$/$C_2H_2$/$C_2H_4$/$C_2H_6$ quaternary mixture[49]. For Zn-fa-datz (1), the narrow cavity ensures that the larger $C_2H_6$ molecule (kinetic diameter = 4.44 Å) can fully contact the pore surface and achieve a higher interaction than the smaller $C_2H_4$ (kinetic diameter = 4.16 Å). Nevertheless, it also causes $C_2H_4$ to bind slightly more strongly to the network than the smaller $CO_2$ (kinetic diameter = 3.30 Å)[53]. We speculated that by fine tuning the pore structure to achieve a more optimized $CO_2$ adsorption environment, it is possible to reverse the adsorption affinity of $C_2H_4$ and $CO_2$ without affecting the adsorption sequence of $C_2H_2$/$C_2H_4$/$C_2H_6$. After carefully analyzing the Zn-fa-datz (1) network, it can be observed that the pore wall of 1D channel is constituted by fa$^{2-}$ ligands and both two amino groups of datz$^-$ ligands through four tight hydrogen-bonding interactions (O··H···N = 1.95–2.12 Å, ∠O-H···N = 138.8–170.4°) (Fig. 2d and Supplementary Fig. 12). The hydrogen bonds restrict the swing of ligands and determine the arrangement of adsorption sites and size/shape of the channel. Hence, we predict that precise pore structure control could be achieved by regulating the hydrogen bonds via different amino side groups (i.e., replacing the diamino datz$^-$ with unilateral-amino 3-amino-1,2,4-triazolate, atz$^-$).

### Synthesis and characterization of Zn-fa-atz (2)

Solvothermal reaction of Zn(NO$_3$)$_2$·6H$_2$O with H$_2$fa and Hatz in a DMF/MeOH/water mixed solvent gave a pillared-layer coordination network with 1D channels, [Zn$_2$(fa)(atz)$_2$] (Zn-fa-atz (2)) (Fig. 2b). Single-crystal structure analysis at 298 K revealed that Zn-fa-atz (2) crystallizes in the orthorhombic *Pbca* space group (Supplementary Table 3), isoreticular with previous Zn-fa-datz (1). Both $Zn^{2+}$ ions in Zn-fa-atz (2) exhibit tetrahedral coordination. Each $Zn^{2+}$ ion is coordinated with three N atoms from three atz$^-$, and one O atom from fa$^{2-}$ (Supplementary Fig. 13). Similar to Zn-fa-datz (1), Zn-fa-atz (2) also exhibits **pcu** topology, which is constructed by the Zn-atz layer based on the dinuclear [Zn$_2$(atz)$_2$] unit and the fa$^{2-}$ pillar, but there is an obvious slip between the pillar and the layer (Supplementary Figs. 1–2). The porosity of

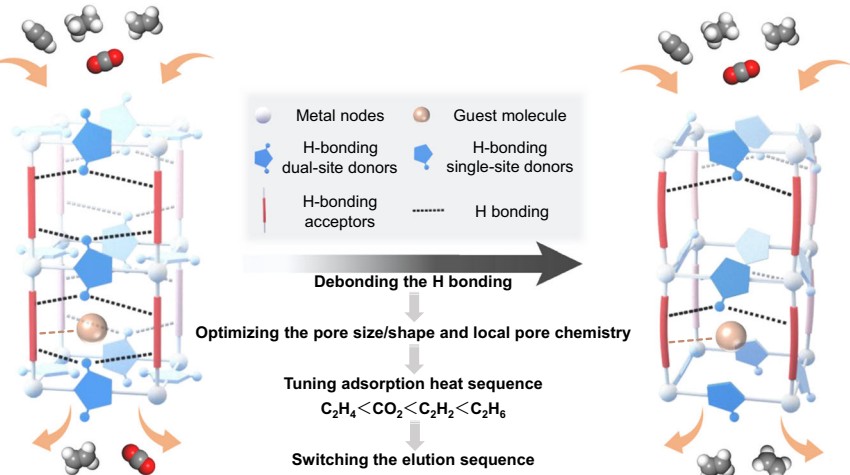

**Fig. 1 | Illustration of strategy.** Illustration of hydrogen bond unlocking-driven pore size/shape and chemistry control to shift multi-component separation (Color code: metal nodes, white; guest molecule, orange; H-bonding single-site/dual-site donors, blue; H-bonding acceptors, red; H bonding, black dotted line; weak interaction, orange dotted line; the direction of gas flow, orange row; derivation of structure-function relationship, black row).

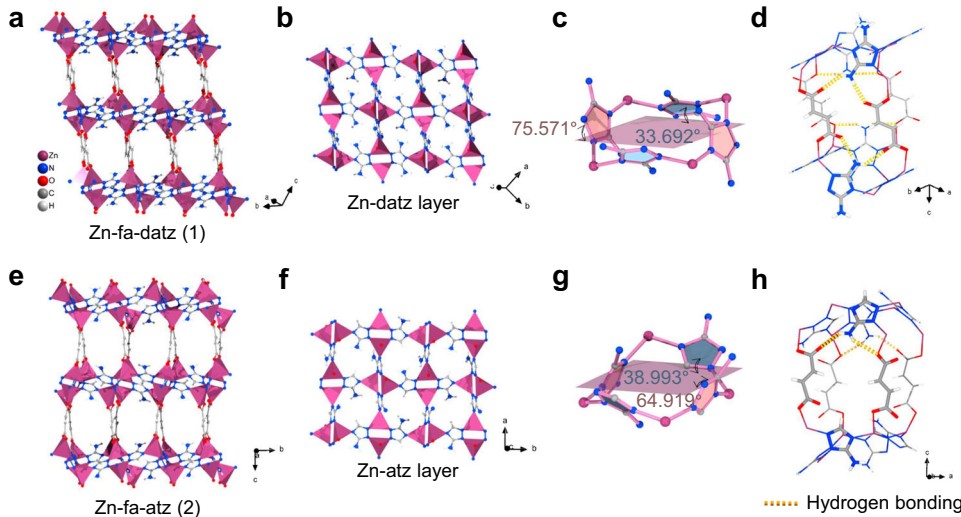

**Fig. 2 | Crystal structures.** Perspective view of the structure along the 1D channels of Zn-fa-datz (1) (**a**) and Zn-fa-atz (2) (**e**). Zinc–aminotriazolate layer of Zn-fa-datz (1) (**b**) and Zn-fa-atz (2) (**f**). Dihedral angles in Zn-fa-datz (1) (**c**) and Zn-fa-atz (2) (**g**) between atz⁻/datz⁻ and Zn-atz/datz layers. Front views of pore walls with highlighted (yellow) H−N···O interactions of Zn-fa-datz (1) (**d**) and Zn-fa-atz (2) (**h**). Color code: Zn, purple; C, gray; O, red; N, blue; H, white.

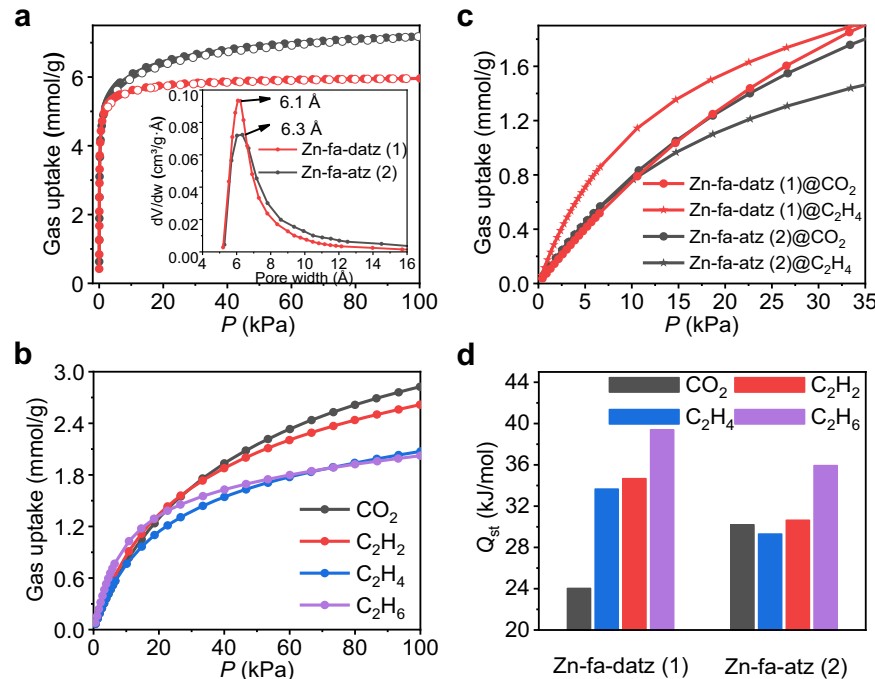

**Fig. 3 | Gas adsorption properties of Zn-fa-datz (1) and Zn-fa-atz (2). a** $CO_2$ adsorption isotherms at 195 K (solid: adsorption; open: desorption) and corresponding pore size distributions calculated based on the $CO_2$ isotherm at 195 K according to the Horvath-Kawazoe model (pore geometry: cylinder) (insert) for Zn-fa-datz (1) (red) and Zn-fa-atz (2) (black). **b** Adsorption isotherms of Zn-fa-atz (2) for $CO_2$ (black), $C_2H_2$ (red), $C_2H_4$ (blue) and $C_2H_6$ (purple) from 0–100 kPa and 298 K. **c** Comparison of $CO_2$ (point) and $C_2H_4$ (star) adsorption isotherms of Zn-fa-datz (1) (red) and Zn-fa-atz (2) (black) from 0 to 35 kPa and 298 K. **d** Comparison of adsorption enthalpies ($Q_{st}$) of four gases in Zn-fa-datz (1) and Zn-fa-atz (2).

Zn-fa-atz (2) was calculated to be 46.3% (by PLATON[54]) which is larger than that of Zn-fa-datz (1) (38.9%) (Supplementary Table 1). Furthermore, due to the reduction of amino groups, there are fewer hydrogen bonds on the pore walls than Zn-fa-datz (1). For Zn-fa-atz (2), only one side of atz⁻ ligands are tied to fa²⁻ ligands through two O-H···N hydrogen bonding interactions (O-H···N = 2.12–2.47 Å, ∠O-H···N = 135.6–170.4°) (Fig. 2h and Supplementary Fig. 12). But in Zn-fa-datz (1), both sides of datz⁻ ligand can connect with fa²⁻ ligands by four hydrogen bonds. When the diagonals between four adjacent O atoms

from different fa²⁻ ligands were used to compare the pore sizes of the two structures (minus the van der Waals radius of O atom of 1.52 Å) (Supplementary Fig. 2)[55], it can be seen that the difference in aperture between Zn-fa-atz (2) (5.5 × 4.9 Å) and Zn-fa-datz (1) (5.4 × 4.6 Å) is very small. In fact, debonding the hydrogen bonds causes the rotation of the five-member ring of atz⁻ ligand, resulting in different dihedral angles between atz⁻/datz⁻ and Zn-atz/datz layers (Fig. 2c, g). Therefore, the greater difference between the Zn-fa-atz (2) and Zn-fa-datz (1) is reflected in the size/shape of the pore and the local pore chemistry.

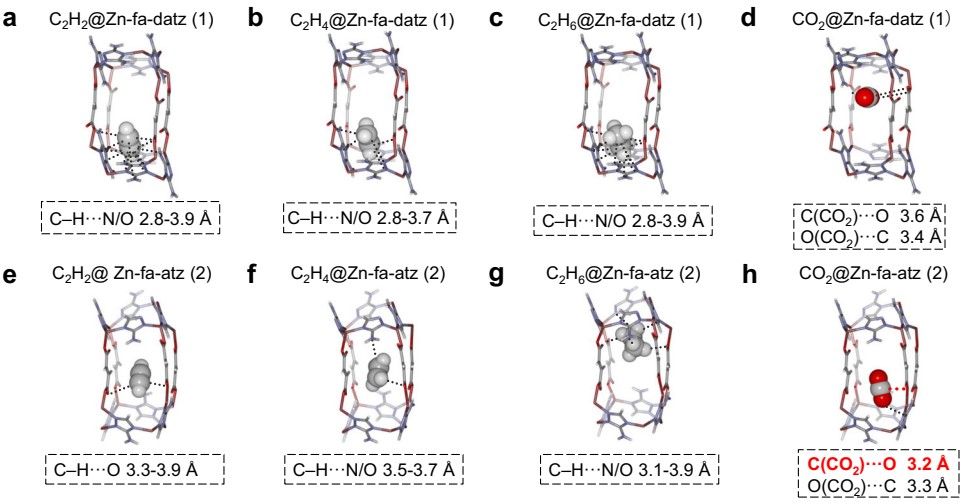

**Fig. 4 | The host-guest structures of Zn-fa-datz (1) and Zn-fa-atz (2) revealed by powder diffraction data from Rietveld refinement analysis. a** The $C_2H_2$ binding site, (**b**) $C_2H_4$ binding site, (**c**) $C_2H_6$ binding site and (**d**) $CO_2$ binding site in gas-loaded Zn-fa-datz (1). **e** The $C_2H_2$ binding site, (**f**) $C_2H_4$ binding site, (**g**) $C_2H_6$ binding site and (**h**) $CO_2$ binding site in gas-loaded Zn-fa-atz (2). Color code: Zn, purple; C, gray; O, red; N, blue; H, white. The graphical representation is created with iRASPA[61].

Thermogravimetry and PXRD data showed that Zn-fa-atz (2) can be fully exchanged with MeOH (Supplementary Figs. 3 and 6). The 195 K $CO_2$ adsorption isotherm for Zn-fa-atz (2) shows a quasi-type-I characteristic. The pore volume was calculated to be 0.285 $cm^3\,g^{-1}$ and 0.283 $cm^3\,g^{-1}$ using the $CO_2$ and $N_2$ uptake measured at $P/P_0 = 0.96$ and 0.95, respectively, which is comparable with the value calculated from single-crystal diffraction data at 195 K (0.319 $cm^3\,g^{-1}$) (Fig. 3a, Supplementary Fig. 14 and Supplementary Table 1). Besides, it is worth to mention that Zn-fa-atz (2) can remain the crystalline and porosity after treated with water or exposed to moisture (*ca.* 35% RH) (Supplementary Fig. 15). The pore size distribution analysis based on the Horvath-Kawazoe model also reveals that the 1D channels of Zn-fa-atz (2) (6.3 Å) comparable with that of Zn-fa-datz (1) (6.1 Å), which is consistent with single-crystal analysis.

## Adsorption and separation performances of Zn-fa-atz (2)

Single-component adsorption isotherms for $CO_2$, $C_2H_2$, $C_2H_4$, and $C_2H_6$ in Zn-fa-atz (2) were measured at 273 K and 298 K (Fig. 3b and Supplementary Fig. 7). At low pressure and 298 K, similar with Zn-fa-datz (1), Zn-fa-atz (2) shows higher uptake for $C_2H_6$ and $C_2H_2$ than $C_2H_4$, indicating the selective adsorption of both adsorbates over $C_2H_4$. However, the $CO_2$ uptake for Zn-fa-atz (2) is higher than that for $C_2H_4$, while the opposite was observed for Zn-fa-datz (1). At 298 K, the adsorption amount sequences of Zn-fa-atz (2) at 14 kPa, 25 kPa and 33 kPa are followed as $C_2H_6 > C_2H_2 > CO_2 > C_2H_4$, $C_2H_2 \approx CO_2 > C_2H_6 > C_2H_4$, and $CO_2 \approx C_2H_2 > C_2H_6 > C_2H_4$, respectively (Supplementary Fig. 16). For Zn-fa-atz (2), the trend in the adsorption enthalpy ($Q_{st}$) at the low loading is as the following: $C_2H_6$ (35.9 $kJ\,mol^{-1}$) > $C_2H_2$ (30.6 $kJ\,mol^{-1}$) > $CO_2$ (30.2 $kJ\,mol^{-1}$) > $C_2H_4$ (29.3 $kJ\,mol^{-1}$) (Fig. 3d, Supplementary Figs. 7–9 and Supplementary Table 2), while the $Q_{st}$ order of Zn-fa-datz (1) is following as $C_2H_6$ (39.4 $kJ\,mol^{-1}$) > $C_2H_2$ (34.7 $kJ\,mol^{-1}$) > $C_2H_4$ (33.6 $kJ\,mol^{-1}$) > $CO_2$ (24.0 $kJ\,mol^{-1}$). Interestingly, when compared with Zn-fa-datz (1), the $C_2H_2/C_2H_4/C_2H_6$ $Q_{st}$ for Zn-fa-atz (2) decreased synchronously and maintained the same sequence, while the $CO_2$ $Q_{st}$ showed a significant increase—that is, Zn-fa-atz (2) reversed the $C_2H_4/CO_2$ adsorption selectivity (Fig. 3c, Supplementary Figs. 10–11 and Supplementary Table 2). Although each of the ideal adsorbed solution theory (IAST) selectivity of the three gases to $C_2H_4$ are not very high ($CO_2/C_2H_4 = 1.4$, $C_2H_2/C_2H_4 = 1.5$, $C_2H_6/C_2H_4 = 1.4$) (Supplementary Table 2), it is rare to achieve the preferential adsorption of $CO_2/C_2H_2/C_2H_6$ over $C_2H_4$ at the same time, especially for components with the

very close physicochemical properties. The IAST selectivities of Zn-fa-datz (1) and Zn-fa-atz (2) were compared with the best-performing sorbents in the $C_2H_2/C_2H_4/C_2H_6$ three-component system, and $C_2H_2/C_2H_4/C_2H_6/CO_2$ four-component system (Supplementary Table 4). Both Zn-fa-datz (1) and Zn-fa-atz (2) show moderate selectivity for $C_2H_2/C_2H_4$, $C_2H_6/C_2H_4$ and $CO_2/C_2H_4$. In fact, it is very difficult to maintain the lowest selectivity for $C_2H_4$ among the four gas components, because the physicochemical properties of the four gases are too similar. In general, for ultramicropores without specific binding sites (e.g., open-metal sites), the adsorption affinity mainly comes from the combination of various weak interactions (e.g., van der Waals forces and hydrogen bonding) between the guest molecule and the network in multiple orientations. Consequently, even a slight change in the pore size/shape and local pore chemistry can significantly affect the affinity. In case of Zn-fa-datz (1) and Zn-fa-atz (2), based on the SCXRD analysis, different amino groups not only change the local chemical environment of the pore, but also affect hydrogen bonds in the framework which further leads to the change in the size/shape of the channel (the spatial arrangement of the adsorption sites) (Fig. 2 and Supplementary Figs. 2 and 12).

## Host-guest structure studies

To further understand the role of pore structure tuning, the host–guest structures of Zn-fa-datz (1) and Zn-fa-atz (2) were studied by the PXRD analyses and corresponding refinements (Fig. 4, Supplementary Figs. 17–20 and Supplementary Table 5). The eight studied hos–guest systems showed that all the gas molecules preferentially localized within the pockets enclosed by four triazolate moieties and four fa$^{2-}$ ligands. For $C_2H_2/C_2H_4/C_2H_6$, the host–guest interactions are mainly contributed by weak O/N···H–C hydrogen bonding interactions from multiple orientations. $C_2H_2$, $C_2H_4$, and $C_2H_6$ are all confined in the relatively spacious positions in the cavities of Zn-fa-datz (1) and Zn-fa-atz (2), but the molecular orientations change due to the difference in pore shape and pore chemistry. As shown in Fig. 4 and Supplementary Table 5, most measured O/N···H–C distances in Zn-fa-atz (2) are slightly longer than that in Zn-fa-datz (1), which is consistent with the synchronous decrease of the $Q_{st}$ for the three C2 hydrocarbons in Zn-fa-atz (2). For $CO_2$ in Zn-fa-atz (2) (Fig. 4), the optimal position has changed when compared with that in Zn-fa-datz (1), and the O atom from the framework can contact closely with the C atom in $CO_2$, yielding a relative strong

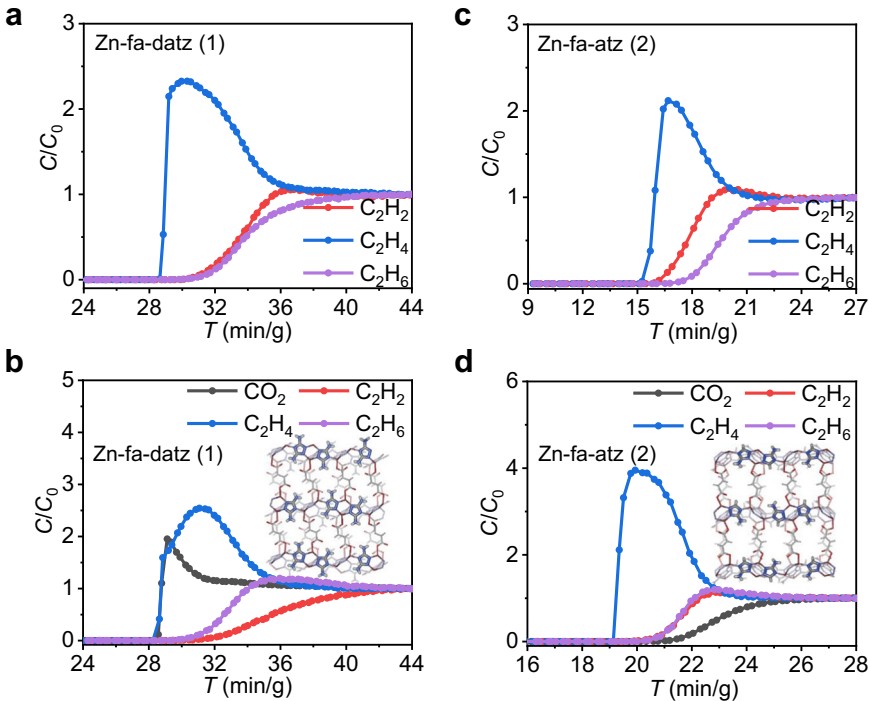

**Fig. 5 | Experimental breakthrough experiments.** Experimental column breakthrough curves at 298 K for: Zn-fa-datz (1) powder using a mixture of (**a**) $C_2H_2$/$C_2H_4$/$C_2H_6$/He (1:1:1:4, total gas pressure of 1 bar, total flowing rate of 3.5 mL/min), and an equimolar mixture of (**b**) $CO_2$/$C_2H_2$/$C_2H_4$/$C_2H_6$ (1:1:1:1, total gas pressure of 1 bar,

total flowing rate of 2 mL/min). Zn-fa-atz (2) powder using (**c**) $C_2H_2$/$C_2H_4$/$C_2H_6$/He (1:1:1:4) and (**d**) $CO_2$/$C_2H_2$/$C_2H_4$/$C_2H_6$ (1:1:1:1). Color code: Zn, purple; C, gray; O, red; N, blue; H, white.

interaction (C···O = 3.231 Å) that cannot be observed in Zn-fa-datz (1), resulting in higher $CO_2$ $Q_{st}$ than that of Zn-fa-datz (1).

## Dynamic breakthrough experiments of Zn-fa-atz (2) and Zn-fa-atz (2)/PES composite beads

To evaluate the gas separation performance, breakthrough experiments were performed with Zn-fa-atz (2). At 298 K and ambient pressure, $C_2H_2$/$C_2H_4$/$C_2H_6$/He and equimolar $CO_2$/$C_2H_2$/$C_2H_4$/$C_2H_6$ mixtures were passed through the column packed with Zn-fa-atz (2) powder (Fig. 5). The outlet gases were monitored by the online gas chromatography. As shown in Fig. 5c, d, $C_2H_4$ breaks through first from the column at 15 min g$^{-1}$ and 19 min g$^{-1}$, respectively, which is consistent with the adsorption isotherms. Before the impurities flowed out, the purity of $C_2H_4$ at the outlet reached that of polymer grade (>99.9%) (Supplementary Fig. 21), demonstrating that Zn-fa-atz (2) achieved the one-step purification of $C_2H_4$ from both the $C_2H_2$/$C_2H_4$/$C_2H_6$/He and $CO_2$/$C_2H_2$/$C_2H_4$/$C_2H_6$ mixtures. It is worth mentioning that the different gas elution orders of three- and four-component separation are related to the order of adsorption capacity of each gas at different partial pressures (Supplementary Table 6). In contrast, Zn-fa-datz (1) can only realize one-step $C_2H_4$ production from ternary $C_2H_2$/$C_2H_4$/$C_2H_6$ mixture. To test the recycling performance of Zn-fa-atz (2), three cycles of breakthrough and following desorption experiments were conducted (Supplementary Figs. 22–23). There was no significant change in the $C_2H_4$ retention time and the separation performance, revealing the favorable recyclability and facile regeneration of Zn-fa-atz (2). Moreover, the highly consistent breakthrough data from three different batches of samples also fully verified the reproducibility of the samples and experiments (Supplementary Fig. 24). The simulated breakthrough curves were conducted to further validate the feasibility of Zn-fa-atz (2) for the separation performance (Supplementary Fig. 25), which is highly consistent with our experimental results. However, when Zn-fa-atz (2) was exposed to wet quaternary mixtures (*ca.* 36% RH), the breakthrough curves showed

that Zn-fa-atz (2) can maintain the elution sequence, but the retention times and the shape of the breakthrough curves have changed significantly, indicating the competitive adsorption of water with the other four gases (Supplementary Fig. 26).

To verify the competitive adsorption during the breakthrough experiments, we calculated the actual uptakes of Zn-fa-datz (1) and Zn-fa-atz (2) for the four gases using the reported method (Supplementary Figs. 27–28 and Supplementary Tables 7–8)[56–58]. The results show that the actual selectivities are little different from that of IAST selectivities. For Zn-fa-atz (2), the adsorption amount order of each gas is followed as $CO_2$ > $C_2H_2$ > $C_2H_6$ > $C_2H_4$, being similar with the isotherms, but the selectivity changed in $CO_2$/$C_2H_4$ ($S_{breakthrough}$ = 2.17 > $S_{IAST}$ = 1.4). For Zn-fa-datz (1), the adsorption amount order for each gas is followed as $C_2H_2$ > $C_2H_6$ > $CO_2$ > $C_2H_4$, which is inconsistent with that of the isotherms ($C_2H_2$ > $C_2H_6$ > $C_2H_4$ > $CO_2$), and the selectivity of $CO_2$/$C_2H_4$ also changed ($S_{breakthrough}$ = 1.27 > $S_{IAST}$ = 0.8). Obviously, both Zn-fa-atz (2) and Zn-fa-datz (1) have different degrees of increase in the adsorption of $CO_2$ in the breakthrough experiments. Therefore, the diffusion coefficients of the four gases through the adsorption kinetic profiles at 298 K (Supplementary Fig. 29) were calculated. The results showed that the diffusion of $CO_2$ (0.3874) was significantly higher than that of $C_2H_4$ (0.1191), $C_2H_2$ (0.0820), and $C_2H_6$ (0.0478), indicating $CO_2$ diffused much faster than other three gases during the breakthrough experiments. Therefore, the larger uptakes of $CO_2$ are the result of the synergistic effect of adsorption thermodynamics and kinetics.

In addition, for practical industrial applications, the Zn-fa-atz (2) crystals were shaped into spherical pellets, with addition of organic polymer binder. In presence of 20 wt% of poly-ether sulfone (PES) as the binder, the Zn-fa-atz (2)/PES composite beads with a diameter of *ca.* 2.5 mm were successfully fabricated via the phase inversion method (Supplementary Fig. 30). The scanning electron microscope (SEM) images show Zn-fa-atz (2) crystals (*ca.* 500 nm) are well embedded within the inner polymer matrix (Supplementary Fig. 30b, c). The $CO_2$ adsorption isotherm at 195 K of Zn-fa-atz (2)/PES also

shows a quasi-type-I characteristic, indicating the microporosity of the beads (Supplementary Figs. 30d, e and 31). The pore volume was calculated to be 0.265 $cm^3 g^{-1}$ at $P/P_0 = 0.96$ (7% lower than that of pure Zn-fa-atz (2) crystal sample), suggesting that Zn-fa-atz (2)/PES retains most of the porosity. Further, the kinetic adsorption profiles for $C_2H_6$ were measured at 298 K and 1 atm. The diffusional rate constants ($k$)[59] for $C_2H_6$ in Zn-fa-atz (2)/PES was calculated to be 1.1437, which is within the vicinity of that for the Zn-fa-atz (2) powder ($k = 1.2706$), meaning that compositing has little effect on the gas diffusion (Supplementary Fig. 32). The equimolar $CO_2/C_2H_2/C_2H_4/C_2H_6$ mixture breakthrough experiment was further tested with Zn-fa-atz (2)/PES beads-packed column at room temperature. As shown in Supplementary Fig. 30f, $C_2H_4$ breakthrough first at 18 min $g^{-1}$, following by $C_2H_6$, $CO_2$, and $C_2H_2$, indicating the effective one-step $C_2H_4$ production ability from quaternary mixture after shaping Zn-fa-atz (2) into PES-based spherical pellets.

## Discussion

In conclusion, fine-tuning pore size/shape and local pore chemistry by regulating the network hydrogen bonding interactions in two related coordination networks can precisely control the adsorption selectivity of $C_2H_4$ in the complex separation systems. The reported ultra-microporous adsorbent, Zn-fa-atz (2), can achieve the effective one-step purification of $C_2H_4$ from $CO_2/C_2H_2/C_2H_4/C_2H_6$ quaternary mixture. Design principle presented here could be helpful to advance the new-generation physisorbent synthesis and application for more complex industry-related separation system.

## Methods

### Synthesis of Zn-fa-datz (1)

According to the reported procedures with a little modification[52]. $Zn(NO_3)_2 \cdot 6H_2O$ (1.0 mmol, 0.298 g), fumaric acid ($H_2$fa, 0.5 mmol, 0.058 g), 1$H$-1,2,4-triazole-3,5-diamino (Hdatz, 1.0 mmol, 0.099 g) were dissolved in 10 mL DMF/MeOH/$H_2O$ mixed solution (4:4:2, $v/v/v$). After 30 min of sonication treatment, the resulting solution was sealed in a 25 mL Teflon-lined stainless-steel autoclave and heated at 130 °C under autogenous pressure for 72 h. After slowly cooling down to room temperature, the colorless crystals of Zn-fa-datz (1) were washed with DMF/MeOH (1:1, $v/v$) mixed solution for three times, and dried at room temperature (yield = 48% based on Zn). The obtained sample was exchanged with fresh MeOH three times daily for three days.

### Synthesis of Zn-fa-atz (2)

$Zn(NO_3)_2 \cdot 6H_2O$ (1.0 mmol, 0.298 g), fumaric acid ($H_2$fa, 0.5 mmol, 0.058 g), 3-amino-1,2,4-triazole (Hatz, 1.0 mmol, 0.084 g) were dissolved in 10 mL DMF/MeOH/$H_2O$ mixed solution (4:4:2, $v/v/v$). After 30 min of sonication, the resulting solution was sealed in a 25 mL Teflon-lined stainless-steel autoclave and heated at 130 °C under autogenous pressure for 72 h. After slowly cooling down to room temperature, the colourless crystals of Zn-fa-atz (2) were washed with fresh DMF/MeOH (1:1, $v/v$) mixed solution for three times, and dried at room temperature (yield = 52% based on Zn). The obtained sample was exchanged with fresh MeOH three times daily for three days.

### Gas adsorption measurements

The thermodynamic adsorption isotherms for $CO_2$, $C_2H_2$, $C_2H_4$, and $C_2H_6$ were conducted on 3FLEX (Micromeritics). Before the $N_2$ (77 K)/$CO_2$ (195 K) adsorption measurement, Zn-fa-atz (2) powder, Zn-fa-datz (1) powder and Zn-fa-atz (2)/PES beads (-100 mg) were evacuated under a dynamic vacuum at 75 °C for 4 h to remove the guest molecules.

### Dynamic breakthrough experiments

Before breakthrough experiments, the samples were packed in a column and activated in-situ by heating at 75 °C for 20 h in a He flow with

rate of 20 mL/min, and then cooled to room temperature. Then the gas flow is switched to the desired gas mixture ($v$(He)/$v$($C_2H_2$)/$v$($C_2H_4$)/$v$($C_2H_6$) = 58:14:14:14, $v$($CO_2$)/$v$($C_2H_2$)/$v$($C_2H_4$)/$v$($C_2H_6$) = 25:25:25:25). The dynamic breakthrough data were recorded on a homemade apparatus at room temperature and 1 atm. The gas stream concentration at column outlet was continuously detected by using a chromatographic analyzer (TCD-Thermal Conductivity Detector, detection limit 0.1 ppm). After equilibrium, desorption curves of Zn-fa-atz (2) in Supplementary Fig. 23 was collected under a He flow of 20 mL/min at 70 °C.

### Reporting summary

Further information on research design is available in the Nature Portfolio Reporting Summary linked to this article.

## Data availability

For full characterization data including detailed sorption and breakthrough experiments data see the Supplementary Methods 3 and 5. All data supporting the finding of this study are available within this article and its Supplementary Information. Crystallographic data for Zn-fa-atz (2) reported in this article have been deposited at the Cambridge Crystallographic Data Centre, under deposition numbers CCDC 2176255-2176256. Copies of the data can be obtained free of charge via https://www.ccdc.cam.ac.uk/structures/. Source data of the PXRD patterns, TGA curves, sorption tests; gas adsorption enthalpies and selectivities, Rietveld refinement of powder X-ray diffraction tests and breakthrough tests that support the findings of this study are provided as a Source Data file (ref. 60. Rong, Y. (2023): Source data of Zn-fa-datz (1) and Zn-fa-atz (2) that support the findings of this study.xlsx. Data sets. figshare https://doi.org/10.6084/m9.figshare.24864540). Source data are provided with this paper.

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

## Acknowledgements

We appreciate the financial support from the National Natural Science Foundation of China (22071195, K.-J. C., 22101231, Y.W.), the Youth Innovation Team of Shaanxi Universities and China Postdoctoral Science Foundation (No. 2022M712585, T.Z.). We are also thankful for the help from Prof. Dr. Bao-Yong Zhu, Prof. Dr. Hui Wang and Mr. You Wang. T.P. and K.A.F. acknowledge the use of services provided by Research Computing at the University of South Florida.

## Author contributions

K.-J.C. designed the project. R.Y., Y.W., J-W.C., T.G., X.-O.X. and Q.-H.Y. synthesized the compounds, J.-W.C., X.J., R.Y., Y.W. and T.G. collected all adsorption data. T.P. and K.A.F. participated in the separation mechanism discussion. R.Y., Y.W. and J-W.C. collected the experimental breakthrough data. R.K. carried out the breakthrough simulation. H.C. and L.Li participated in the breakthrough simulation discussion. Y.W. and T.G. collected the SEM data. R.Y. and Y.W. analyzed the adsorption data and experimental breakthrough data. Y.W., B.-K.L. and T.Z. analyzed the single crystal data, Y.W. and Z.-M.Y. collected and analyzed the powder refinement data. Y.W., R.Y. and K.-J.C. wrote the paper, and all authors contributed to revise the manuscript.

## Competing interests

The authors declare no competing interests.
