## [Peer Review File · Nature Communications]

REVIEWER COMMENTS

Reviewer #1 (Remarks to the Author):

This work by Yang et al. reported two highly-related porous coordination networks (Zn-fa-datz and Zn-fa-atz) for efficient C₂H₄ separation function from C₂H₂/C₂H₄/C₂H₆ ternary mixture and CO₂/C₂H₂/C₂H₄/C₂H₆ quaternary mixture system. Their results revealed that reducing amino groups on the organic ligand resulted in less hydrogen bonds in the host network, thus fine-tuning the pore size to achieve the effective one-step purification of C₂H₄ from quaternary mixture. The separation performance were confirmed by gas sorption isotherms and dynamic breakthrough experiments, and the separation mechanism was studied by GCMC simulations. However, the separation performance have no obvious improvement compared with their previous similar work (Nat. Commun., 2021, 12, 6507). The mechanism is not clear in this work that cannot support their main claims, and the reported mechanism by GCMC simulations is not reliable and solid, especially for the weak host-guest interactions. Therefore, I cannot recommend its publication in Nat. Commun. Below are some detailed concerns.

- (1) The surface area and pore volume were determined by 195 K CO₂ sorption isotherms. Why not use 77 K N₂ isotherms. And the results are suggested to compare with the theoretical values obtained by single-crystal analysis.
- (2) The pore sizes revealed in Fig. 3a are very similar (6.1 and 6.3 Å) for Zn-fa-datz and Zn-fa-atz, which are both much larger than the molecular sizes of all the C₂ and CO₂ molecules. It seems that pore size control might be not the key factor to the gas sorption changes between the two materials.
- (3) The reported mechanism by GCMC simulations is not reliable and solid to support their claims, especially for the weak host-guest interactions. So the separation mechanism is not clear in this work, and the authors are suggested to provide more reliable evidence, such as single-crystal X-ray diffraction studies on gas-loaded structures.
- (4) The chemical stabilities of these materials toward moisture and water are suggested to be examined, which is important to practical industrial applications.
- (5) The dynamic breakthrough experiments under humidity conditions are also need to be studied in detail.

Reviewer #2 (Remarks to the Author):

Yang et al., present a nice work for the precise pore size control of MOF by regulating the framework hydrogen bond toward complex separation systems. Rational modulation of pore structure is regularly attempted in two- or three-component separation systems; however, this is the first example that is available for the quaternary mixture systems. Obviously, the increase of components will greatly increase the difficulty of separation. This work demonstrates the means of precise control in maneuvering pore sizes at sub-angstrom precision by tailoring the network hydrogen bonds and elucidates the impact of the slight change in pore size on selectivity in a complex system. Impressively, the new MOF can still show a good ability to separate C₂H₄ in one step even in such a complex mixture system, which I thought was very rare in the current research background. Besides, the molded MOF can also maintain

good performance, which brings the result closer to industrial application. I believe that the topic and approach described in this manuscript are of general interest to a broad audience. I thus strongly recommend its publication in *Nat. Commun.*, with a few minor points being addressed:

1. In this manuscript, the author calculates the IAST selectivity of the compound and states that the selectivity is not very high. I also believe that in such a complex mixture system, the comparison of IAST selectivity is not dominant, but I still suggest that the authors add IAST data of related compounds in three-component and four-component systems, which will give readers a more intuitive understanding of this separation field.
2. The hydrogen bond is the reason for the slight pore size difference between Zn-fa-datz (1) and Zn-fa-atz (2), but the related parameters of the framework hydrogen bond in the two structures do not seem to be mentioned in this paper. The manuscript or SI should reflect hydrogen bond parameters and related structural details.
3. The interaction distances between the four guests and the framework are mentioned in Fig. 4. Although hydrogen bonding and electrostatic interactions usually come from multiple directions and are difficult to count for ultramicroporous structures without specific adsorption sites, as a scientific paper, I suggest that clearer interaction distances and angles in a reasonable range (e.g., $< 4 \text{ \AA}$ and $> 120^\circ$ for hydrogen bonds) can be reflected in SI in the form of figures and tables.
4. The authors use the LDF model to calculate and compare the diffusion coefficients of C_2H_6 in Zn-fa-atz (2) and Zn-fa-atz (2)/PES, but the related calculation methods are not mentioned. Please describe the method.
5. The names of all compounds in the figures should be bold to correspond to the article.
6. Some numerical values are mentioned, but there are no corresponding references.
(1) In the process of ethylene (C_2H_4) production, the product compositions of the thermal decomposition are complicated, and the conversion of dehydrogenation is only ca. 50%–60%.
(2) However, limited by the physicochemical properties of four gas molecules (kinetic diameter: $\text{CO}_2 \approx \text{C}_2\text{H}_2 < \text{C}_2\text{H}_4 < \text{C}_2\text{H}_6$; quadruple moment: $\text{C}_2\text{H}_2 > \text{CO}_2 > \text{C}_2\text{H}_4 > \text{C}_2\text{H}_6$), it is extremely difficult to separate C_2H_4 from the quaternary $\text{CO}_2/\text{C}_2\text{H}_2/\text{C}_2\text{H}_4/\text{C}_2\text{H}_6$ in one step.

Reviewer #3 (Remarks to the Author):

This manuscript handles the selective separation of ethylene from multi-component mixtures by tuning the pore control of Zn-fa-atz. As compared with Zn-fa-datz, Zn-fa-atz has relatively larger pores by fewer hydrogen bonds in the host network, as the authors mentioned. Based on single or mixed gas experiments, it is clear that Zn-fa-atz has more affinity with CO_2 , and less affinity with ethylene, resulting in more efficient C_2H_4 separation. The results sound great, but this manuscript lacks a discussion. Particularly, there is no discussion of why CO_2 affinity was improved in Zn-fa-atz but C_2H_4 affinity was reduced in detail. The molecular simulations explained them, but molecular simulation results cannot elucidate some possible competitive sorption in the mixed gas. For publication, in my opinion, the discussion section should be strengthened.

Reviewer #1:

This work by Yang et al. reported two highly-related porous coordination networks (Zn-fa-datz and Zn-fa-atz) for efficient C₂H₄ separation function from C₂H₂/C₂H₄/C₂H₆ ternary mixture and CO₂/C₂H₂/C₂H₄/C₂H₆ quaternary mixture system. Their results revealed that reducing amino groups on the organic ligand resulted in less hydrogen bonds in the host network, thus fine-tuning the pore size to achieve the effective one-step purification of C₂H₄ from quaternary mixture. The separation performance were confirmed by gas sorption isotherms and dynamic breakthrough experiments, and the separation mechanism was studied by GCMC simulations. However, the separation performance have no obvious improvement compared with their previous similar work (Nat. Commun., 2021, 12, 6507). The mechanism is not clear in this work that cannot support their main claims, and the reported mechanism by GCMC simulations is not reliable and solid, especially for the weak host-guest interactions. Therefore, I cannot recommend its publication in Nat. Commun. Below are some detailed concerns.

(1) The surface area and pore volume were determined by 195 K CO₂ sorption isotherms. Why not use 77 K N₂ isotherms. And the results are suggested to compare with the theoretical values obtained by single-crystal analysis.

Response: Many thanks for the reviewer's suggestion. We agree with the reviewer that it is a consensus in the field to use the 77 K N₂ sorption isotherms to calculate pore volume. Thus, the 77 K N₂ sorption isotherms were also measured before our submission. However, during the measurement we found that the narrower channels of **Zn-fa-datz (1)** make it harder for the N₂ (larger kinetic size) diffusion, resulting in an extremely long equilibrium time for the isotherm measurement (only four data points were collected after 28.8 hours, and the normal cooling dewar bottle can not support the accomplishment of such a full test) (**Figure R1**). As shown in **Supplementary Figure 4** of the revised version, the kinetic data of **Zn-fa-datz (1)** for N₂ and CO₂ were recorded at 77 K and 195 K, respectively. The diffusional rate constant of CO₂ (0.8883) is nearly 52 times as much as that of N₂ (0.0171) for **Zn-fa-datz (1)**. Therefore, for the sake of comparison, the surface area and pore volume were determined by 195 K CO₂ sorption isotherms.

As shown in **Supplementary Figure 14** and **Table 1** of the revised version, for **Zn-fa-atz (2)**, the measured pore volume values calculated by 195 K CO₂ (0.285 cm³ g⁻¹) and 77 K N₂ (0.283 cm³ g⁻¹) are almost the same, indicating that 77 K N₂ isotherm is equivalent to 195 K CO₂ isotherm to characterize the permanent porosity of **Zn-fa-atz (2)**.

Accordingly, we added the statement “The purity and porosity were confirmed by powder X-ray diffraction (PXRD) pattern and 195 K CO₂ adsorption isotherm, respectively (Fig. 3a, Supplementary Fig. 3 and Table 1). Note that, Because N₂ diffuses extremely slowly in **Zn-fa-datz (1)** (Supplementary Fig. 4), 195 K CO₂ adsorption isotherm was conducted for the study of the porosity” to the revised main text.

“The pore volume was calculated to be 0.285 cm³ g⁻¹ using the CO₂ uptake measured at $P/P_0 = 0.96$, which is comparable with the value calculated from single-crystal diffraction data at 195 K (0.319 cm³ g⁻¹) (Fig. 3a and Supplementary Table 1).” was revised to “The pore volume was calculated to be 0.285 cm³ g⁻¹ and 0.283 cm³ g⁻¹ using the CO₂ and N₂ uptake measured at $P/P_0 = 0.96$ and 0.95, respectively, which is comparable with the value calculated from single-crystal diffraction data at 195 K (0.319 cm³ g⁻¹) (Fig. 3a, Supplementary Fig. 14 and Supplementary Table 1).”

Figure R1. N₂ and CO₂ adsorption isotherms of **Zn-fa-datz (1)** collected in 10.06 and 28.8 hours at 77 K and 195 K, respectively.

Supplementary Figure 4. Diffusional rate constants of N₂ (77 K) and CO₂ (195 K) in **Zn-fa-datz (1)**.

Supplementary Figure 14. Comparison of the adsorption isotherms of 77 K N₂ (red) and 195 K CO₂ (black) for **Zn-fa-atz (2)**.

Supplementary Table 1. Summary of the structural information, adsorption data of **Zn-fa-datx (1)** and **Zn-fa-atz (2)**.

Compound	Zn-fa-atz (2)			Zn-fa-datx (1)	
	Temperature	298 K	195 K	77 K	293 K
Topology	pcu			pcu	
Pore size ^[a]		6.3			6.1
Density (g cm ⁻³)	1.284	1.336		1.428	
Volume ratio ^[b]	46.3	42.6		38.9	
Pore volume (measured) ^[c]	0.361	0.319		0.272	
Pore volume (measured from the uptake of the CO ₂ isotherm) ^[d]		0.285			0.237
Pore volume (measured from the uptake of the N ₂ isotherm) ^[e]			0.283		

[a] Pore size (Å) calculated based on the CO₂ isotherm at 195 K according to the Horvath-Kawazoe model (pore geometry: cylinder). [b] Volume ratio (%) estimated by PLATON software¹ without consideration of the solvent in the pore. [c] The calculated pore volume (cm³ g⁻¹) estimated from single crystal data using PLATON software. [d] The measured pore volume (cm³ g⁻¹) calculated from the uptake at $P/P_0 = 0.96$ assuming the liquid CO₂ filling in the pore. [e] The measured pore volume (cm³ g⁻¹) calculated from the uptake of the N₂ isotherm at 77 K assuming the liquid N₂ filling in the pore, at $P/P_0 = 0.95$ for **Zn-fa-atz (2)**.

(2) The pore sizes revealed in Fig. 3a are very similar (6.1 and 6.3 Å) for Zn-fa-datz and Zn-fa-atz, which are both much larger than the molecular sizes of all the C₂ and CO₂ molecules. It seems that pore size control might be not the key factor to the gas sorption changes between the two materials.

Response: Many thanks for the reviewer's suggestion. We do agree with the reviewer that the small transformation of the aperture can hardly affect the affinity with guests when the pore size of the framework is larger than the guest molecular size. We realize that the original description of the pore size is not rigorous enough, so we made corresponding revisions and also the title of the manuscript.

In fact, for the ultramicroporous structure, it is difficult to describe the interactions between the framework and the guest molecules through a simple visualization concept, because it must have sites oriented from multiple directions. In this work, the differences of Q_{st} between the four gases we studied are already very small, which is also the key challenge in multi-component separation systems. In this case, only a very small difference will change the adsorption order in the breakthrough experiment, and this requires the control of the pore structure to be very accurate.

In ultramicroporous **Zn-fa-datz (1)** and **Zn-fa-atz (2)**, although there is a difference in the number of -NH₂ groups between **Zn-fa-datz (1)** and **Zn-fa-atz (2)**, the -NH₂ groups of both two structures are actually hidden in the frameworks, which makes it difficult to interact with the guests. Based on the SCXRD analysis, the biggest change caused by amino groups is the number of hydrogen bonds in the framework which further influenced the pore structure (including pore size and pore chemistry) (**Figure 4** and **Supplementary Figure 12**). This influence not only changes the pore sizes but also the spatial arrangement of the adsorption sites in the channel.

In order to clarify the separation mechanism more clearly, we calculated the actual adsorption amount of the adsorbent to each component gas in the breakthrough experiment (**Supplementary Figures 34-35** and **Tables 10-11**). The results show that the actual selectivities are little different from that of IAST selectivities. For **Zn-fa-atz (2)**, the adsorption amount order of each gas is followed as CO₂ > C₂H₂ > C₂H₆ > C₂H₄, being similar with the isotherms, but the selectivity changed in CO₂/C₂H₄ ($S_{breakthrough}=2.17 > S_{IAST}=1.4$). For **Zn-fa-datz (1)**, the adsorption amount order for each gas is followed as C₂H₂ > C₂H₆ > CO₂ > C₂H₄, which is inconsistent with that of the isotherms (C₂H₂ > C₂H₆ > C₂H₄ > CO₂), and the selectivity of CO₂/C₂H₄ also changed ($S_{breakthrough}=1.27 > S_{IAST}=0.8$). Obviously, both **Zn-fa-**

atz (2) and **Zn-fa-datz (1)** have different degrees of increase in the adsorption of CO₂ in the breakthrough experiments.

We considered that this phenomenon is reasonable, because there are many influencing factors that cannot be reflected by the single-component isotherm in the actual separation scenario, such as the interaction between guest-guest interaction in the framework at the high-pressure section, and the adsorption kinetics of different molecules in mixed systems. Firstly, from a thermodynamic point of view, as shown in **Supplementary Figure 9**, at the low-loading section, the Q_{st} order of adsorption of the four gases by **Zn-fa-atz (1)** is C₂H₆>C₂H₂>CO₂>C₂H₄, while the order change to be C₂H₆≈CO₂>C₂H₂>C₂H₄ when the loading reaches 2 mmol/g. The Q_{st} of CO₂ does not change obviously with the increase of loading, while the other three gases have a decreasing trend, which is probably caused by the strong guest-guest interaction of CO₂. This phenomenon has been confirmed by the single crystal structure in the previous study. In the host-guest structure, with the increase of pressure, the C and O atoms of different CO₂ molecules can contact each other through electrostatic attraction, resulting in no significant decrease in the heat of adsorption at high loading (**Figure R2**, ref. *Science*, **2010**, 330, 650). This is only the result inferred from the single-component isotherms, and the adsorption process of the mixture should be more complicated. Moreover, to verify the influence of adsorption kinetics, the diffusion time coefficients (D') of CO₂/C₂H₂/C₂H₄/C₂H₆ in **Zn-fa-atz (1)** were calculated from the adsorption kinetic profiles measured at 298 K. As shown in **Supplementary Figure 36**, the D' value sequence of the four gases is followed as CO₂ (0.3874) >> C₂H₄ (0.1191) >C₂H₂ (0.0820) >C₂H₆ (0.0478), indicating CO₂ diffused much faster than other three gases during the breakthrough experiments. Although the Q_{st} of CO₂ is slightly lower than that of C₂H₆, the large diffusion difference causes CO₂ to be adsorbed faster in the breakthrough experiment. Therefore, the larger uptakes of CO₂ are the result of the synergistic effect of adsorption thermodynamics and kinetics.

These results indicate that there are complex competitive adsorption and co-adsorption behaviors in the breakthrough experiment. Therefore, we very much hope that the reviewer will understand that we cannot give a perfect answer, as it is the most challenging scientific problem in complex separation systems (ref: Materials for Separation Technologies. Energy and Emission Reduction Opportunities, Oak Ridge, TN (United States): Oak Ridge National Lab. (ORNL); **2005**).

Accordingly, the following discussion has been added to the revised main text:

The title “Hydrogen Bond Unlocking-Driven Pore Size Control for Shifting Multi-Component Gas Separation Function” was revised to “Hydrogen Bond Unlocking-Driven Pore **Structure** Control for Shifting Multi-Component Gas Separation Function”.

“Hence, we predict that precise pore size control could be achieved by regulating the hydrogen bonds via reducing the amino side groups (i.e., replacing the diamino datz⁻ with unilateral-amino 3-amino-1,2,4-triazolate, atz⁻)” was revised to “Hence, we predict that precise pore **structure** control could be achieved by regulating the hydrogen bonds via reducing the amino side groups (i.e., replacing the diamino datz⁻ with unilateral-amino 3-amino-1,2,4-triazolate, atz⁻)”

“Consequently, even a slight change in pore size can significantly affect the affinity, especially for guests with larger molecular sizes.² In case of **Zn-fa-datz (1)** and **Zn-fa-atz (2)**, it can be inferred that the smaller C₂H₂/C₂H₄/C₂H₆ adsorption enthalpy of **Zn-fa-atz (2)** is the outcome of the larger pore size.” was revised to “Consequently, even a slight change in pore structure can significantly affect the affinity, especially for guests with larger molecular sizes. In case of **Zn-fa-datz (1)** and **Zn-fa-atz (2)**, although there is a difference in the number of -NH₂ groups between **Zn-fa-datz (1)** and **Zn-fa-atz (2)**, the -NH₂ groups of both two structures are actually hidden in the frameworks, which makes it difficult to interact with the guests. Based on the SCXRD analysis, the biggest change caused by amino groups is still the number of hydrogen bonds in the framework which further influenced the pore structure (including pore size and pore chemistry) (Figure 4 and Supplementary Fig. 12). This influence not only changes the pore size but also the spatial arrangement of the adsorption sites in the channel.”

“Because of the very small sizes relative to the aperture, C₂H₂, C₂H₄, and C₂H₆ are all confined in the relatively spacious positions in the cavities of **Zn-fa-datz (1)** and **Zn-fa-atz (2)**, but the molecular orientations change due to the slight difference in pore size.” was revised to “Because of the very small sizes relative to the aperture, C₂H₂, C₂H₄, and C₂H₆ are all confined in the relatively spacious positions in the cavities of **Zn-fa-datz (1)** and **Zn-fa-atz (2)**, but the molecular orientations change due to the slight difference in pore **structure**.”

“Due to the smaller size than that of the aperture, CO₂ can freely select the optimal positions to fully contact the networks. Therefore, the increase in pore size leads to a dramatic change of the CO₂ molecule location.” was revised to “Due to the **spatial arrangement of the adsorption sites N/O atoms in the framework**, the optimal position of CO₂ was dramatic changed.”

“To verify the competitive adsorption during the breakthrough experiments, we calculated the actual uptakes of **Zn-fa-datz (1)** and **Zn-fa-atz (2)** for the four gases using the reported method (Supplementary Figs. 34-35 and Tables 10-11) [ref: *Angew. Chem. Int. Ed.* **2019**, *58*, 7692-7696; *Angew. Chem. Int. Ed.* **2020**, *59*, 23322-23328; *Chem* **2021**, *7*, 1006-1019]. The results

show that the actual selectivities are little different from that of IAST selectivities. For **Zn-fa-atz (2)**, the adsorption amount order of each gas is followed as $\text{CO}_2 > \text{C}_2\text{H}_2 > \text{C}_2\text{H}_6 > \text{C}_2\text{H}_4$, being similar with the isotherms, but the selectivity changed in $\text{CO}_2/\text{C}_2\text{H}_4$ ($S_{\text{breakthrough}}=2.17 > S_{\text{IAST}}=1.4$). For **Zn-fa-datz (1)**, the adsorption amount order for each gas is followed as $\text{C}_2\text{H}_2 > \text{C}_2\text{H}_6 > \text{CO}_2 > \text{C}_2\text{H}_4$, which is inconsistent with that of the isotherms ($\text{C}_2\text{H}_2 > \text{C}_2\text{H}_6 > \text{C}_2\text{H}_4 > \text{CO}_2$), and the selectivity of $\text{CO}_2/\text{C}_2\text{H}_4$ also changed ($S_{\text{breakthrough}}=1.27 > S_{\text{IAST}}=0.8$). Obviously, both **Zn-fa-atz (2)** and **Zn-fa-datz (1)** have different degrees of increase in the adsorption of CO_2 in the breakthrough experiments. Therefore, the diffusion coefficients of the four gases through the adsorption kinetic profiles at 298 K (Supplementary Fig. 36) were calculated. The results showed that the diffusion of CO_2 (0.3874) was significantly higher than that of C_2H_4 (0.1191), C_2H_2 (0.0820), and C_2H_6 (0.0478), indicating CO_2 diffused much faster than other three gases during the breakthrough experiments. Therefore, the larger uptakes of CO_2 are the result of the synergistic effect of adsorption thermodynamics and kinetics.”

“In conclusion, fine-tuning of pore size by regulating the network hydrogen bonding interactions in two related coordination networks can precisely control the adsorption selectivity of C_2H_4 in the complex separation systems.” was revised to “In conclusion, fine-tuning of pore **structure** by regulating the network hydrogen bonding interactions in two related coordination networks can precisely control the adsorption selectivity of C_2H_4 in the complex separation systems.”

Figure 4. GCMC adsorption simulation and adsorbed structures. Preferential $\text{C}_2\text{H}_2/\text{C}_2\text{H}_4/\text{C}_2\text{H}_6/\text{CO}_2$ sorption sites in **Zn-fa-datz (1)** and **Zn-fa-atz (2)** obtained by GCMC simulations. Atom colors: C, gray; H, white; N, blue; O, red; Zn, purple.

Supplementary Figure 12. The hydrogen bond parameters of **Zn-fa-datz (1)** and **Zn-fa-atz (2)**.

Supplementary Figure 34. Experimental breakthrough curves for He of 2 mL/min **Zn-fa-atz (2)** at room temperature and 1 bar.

Supplementary Figure 35. Experimental breakthrough curves of **Zn-fa-datz (1)** and **Zn-fa-atz (2)** for quaternary CO₂/C₂H₂/C₂H₄/C₂H₆ mixtures (1:1:1:1, v/v/v/v) with velocity correction.

Supplementary Figure 9. Gas adsorption enthalpies of **Zn-fa-datz (1)** (a), **Zn-fa-atz (2)** (b) calculated by virial method.

Figure R2. Direct observation of the guest-guest interaction between two different CO_2 molecules in a microporous MOF by single-crystal X-ray analysis. Adopted from ref. *Science*, **2010**, 330, 650.

Supplementary Figure 36. Adsorption kinetics profiles (point) and linear fittings (line) of CO_2 (black), C_2H_2 (red), C_2H_4 (blue) and C_2H_6 (green) for **Zn-fa-atz (2)** at 298 K.

Supplementary Table 10. The $\text{CO}_2/\text{C}_2\text{H}_2/\text{C}_2\text{H}_4/\text{C}_2\text{H}_6$ breakthrough performances of **Zn-fa-datz (1)**.

Zn-fa-datz (1)	CO_2 uptake (mmol/g)	C_2H_2 uptake (mmol/g)	C_2H_4 uptake (mmol/g)	C_2H_6 uptake (mmol/g)	S ($\text{C}_2\text{H}_6/\text{C}_2\text{H}_4$)	S ($\text{C}_2\text{H}_2/\text{C}_2\text{H}_4$)	S ($\text{CO}_2/\text{C}_2\text{H}_4$)
Adsorption isotherm at 0.25 bar (298 K)	1.54	2.31	1.69	1.97	-	-	-
Breakthrough experiment	0.52	0.73	0.41	0.63	1.6	1.6	0.8
IAST	-	-	-	-	1.54	1.78	1.27

[a] The single-component gas adsorption uptake data calculated by dual-site Langmuir-Freundlich fittings at partial pressure of 0.25 bar for C_2 gases.

Supplementary Table 11. The CO₂/C₂H₂/C₂H₄/C₂H₆ breakthrough performances of **Zn-fa-atz (2)**.

Zn-fa-atz (2)	CO ₂ uptake (mmol/g)	C ₂ H ₂ uptake (mmol/g)	C ₂ H ₄ uptake (mmol/g)	C ₂ H ₆ uptake (mmol/g)	S (C ₂ H ₆ /C ₂ H ₄)	S (C ₂ H ₂ /C ₂ H ₄)	S (CO ₂ /C ₂ H ₄)
Adsorption isotherm at 0.25 bar (298 K)	1.5	1.5	1.3	1.4	-	-	-
Breakthrough experiment at 1 bar (298 K)	0.50	0.46	0.23	0.43	1.87	2.00	2.17
IAST	-	-	-	-	1.4	1.5	1.4

(3) The reported mechanism by GCMC simulations is not reliable and solid to support their claims, especially for the weak host-guest interactions. So the separation mechanism is not clear in this work, and the authors are suggested to provide more reliable evidence, such as single-crystal X-ray diffraction studies on gas-loaded structures.

Response: Many thanks for the reviewer's suggestion. We agree with the reviewer that single-crystal X-ray diffraction studies on gas-loaded structures are more reliable evidence than that of computer simulations, and it will undoubtedly improve the quality of our work. Therefore, we tried our best to obtain the single crystal gas-loaded structure in last five months. However, it was difficult to retain sample single-crystallinity after a series of post-synthetic treatments (exchanging, heating, adsorption, etc.) because the single crystals of **Zn-fa-atz (2)** are brittle flakes (**Figure R3**). In order to get experimental evidence, PXRD analyses were conducted on gas-loaded **Zn-fa-datz (1)** and **Zn-fa-atz (2)** followed by Rietveld structural refinements to unveil the binding sites of C2 hydrocarbons and CO₂.

As shown in **Supplementary Figure 25** and **Table 8** of the revised version, all these gas molecules are adsorbed near the pockets enclosed by four triazolate moieties and four fa²⁻ ligands, which is similar with the host-guest structures obtained from GCMC simulations, but there are some differences in structural details. This may be because the GCMC structures present the first adsorption site, while the experimental structures are periodic and obtained at high gas pressure. As shown in **Supplementary Table 8**, for C2 hydrocarbon guest molecules, it is reasonable that the **Zn-fa-datz (1)** with smaller pore size has a closer interaction with the guests including C₂H₄, which is why all three hydrocarbon molecules have the decreased adsorption energy in **Zn-fa-atz (2)**. For CO₂ in **Zn-fa-atz (2)**, the optimal position has changed, and the O atom from the framework and the C atom in CO₂ yield a relative strong interaction that cannot be observed in **Zn-fa-datz (1)**, resulting in higher CO₂ Q_{st} than that of **Zn-fa-datz (1)**.

Accordingly, the following discussion has been added to the revised main text:

“Additionally, the gas-loaded structures were also studied by the PXRD analyses and corresponding refinements (Supplementary Fig. 25-29 and Table 8). The results are similar with the host-guest structures obtained from GCMC simulations, but there are some differences in structural details. This may be because the simulated structures present the first adsorption site, while the experimental structures are periodic and obtained at high gas pressure. For C₂ hydrocarbon guest molecules, it is reasonable that the **Zn-fa-datz (1)** with smaller pore size has a closer interaction with three C₂ hydrocarbon molecules. For CO₂ in **Zn-fa-atz (2)**, the optimal position has changed, and the O atom from the framework can contact closely with the C atom in CO₂, yielding a relative strong interaction (C···O = 3.231 Å) that cannot be observed in **Zn-fa-datz (1)**, resulting in higher CO₂ Q_{st} of **Zn-fa-atz (2)** than that of **Zn-fa-datz (1)**.”

Details of PXRD analyses on gas-loaded structures were added to supplementary “Pawley and Rietveld refinement of PXRD” section of the SI, and the host-guest structures and related refinement parameters have been listed in **Supplementary Figure 25-29** of the SI, all of which highlighted in red.

Figure R3. Optical microscope image of **Zn-fa-atz (2)** crystal.

Supplementary Figure 25. (Top) the host-guest structures of **Zn-fa-datz (1)** and **Zn-fa-atz (2)** revealed by powder diffraction data from Rietveld refinement analysis; (down) magnified CO₂-host structures by powder diffraction data from Rietveld refinement analysis.

Supplementary Figure 26. Rietveld refinement plots of powder X-ray diffraction data of CO₂-loaded **Zn-fa-datz (1)** and C₂H₂-loaded **Zn-fa-datz (1)**.

Supplementary Figure 27. Rietveld refinement plots of powder X-ray diffraction data of C₂H₄-loaded **Zn-fa-datz (1)** and C₂H₆-loaded **Zn-fa-datz (1)**.

Supplementary Figure 28. Rietveld refinement plots of powder X-ray diffraction data of CO₂-loaded **Zn-fa-atz (2)** and C₂H₂-loaded **Zn-fa-atz (2)**.

Supplementary Figure 29. Rietveld refinement plots of powder X-ray diffraction data of C₂H₄-loaded **Zn-fa-atz (2)** and C₂H₆-loaded **Zn-fa-atz (2)**.

Supplementary Methods 9. Pawley and Rietveld refinement of PXRD

The microcrystalline **Zn-fa-datz (1)** and **Zn-fa-atz (2)** was placed in a glass capillary ($\Phi = 0.8 \text{ mm}$) connected with an automatic volumetric sorption apparatus (Micromeritics 3FLEX), and heated under high vacuum at 75 °C for 4 hours. After that, CO₂, C₂H₂, C₂H₄ and C₂H₆ gas was introduced by cooling the samples with dry ice-acetone bath to 195 K, and the gas dosed volumetrically from calibrated pressure. PXRD data of the gas-loaded samples was collected on a Rigaku SmartLab X-ray powder diffractometer (Cu K α) with a scanning speed of 0.01 °/step and 7 s/step under capillary transmission mode. All the indexing and refinement were performed by the Reflex plus module of Material Studio 5.0. The pseudo-Voigt profile parameters, background parameters, the cell parameters, the zero point of the diffraction pattern, the global isotropic atom displacement parameters, the Berar-Baldinozzi asymmetry correction parameters, and the March-Dollase preferred orientation correction parameters were optimized

step by step to improve the agreement between the calculated and the experimental powder diffraction patterns.

Supplementary Table 8. Host-guest interactions in PXRD analyses and structural refinements of **Zn-fa-datz (1)** and **Zn-fa-atz (2)** loaded with CO₂, C₂H₂, C₂H₄ and C₂H₆.

Guest molecules	Zn-fa-datz (1)			Zn-fa-atz (2)				
		H···A (Å)	D-H···A (°)	D···A (Å)		H···A (Å)	D-H···A (°)	D···A (Å)
C ₂ H ₂	C51-H53···N48	2.818	163.268	3.856	C24-H26···N94	3.852	121.854	4.499
	C51-H53···N7	3.825	132.105	4.612				
C ₂ H ₄	C50-H53···N44	1.829	164.048	2.895	C93-H96···N125	3.022	126.05	3.747
	C50-H53···N43	2.251	131.006	3.080	C93-H96···O45	3.277	160.87	4.295
	C50-H53···N45	2.564	134.221	3.416	C92-H95···O3	3.507	121.13	4.157
	C50-H52···N21	2.651	163.689	3.712	C92-H94···O46	3.545	149.50	4.516
	C50-H53···N67	2.682	126.992	3.451	C92-H95···N101	3.584	168.60	4.630
	C50-H53···N48	3.148	127.414	3.909	C93-H97···O67	3.987	124.60	4.691
C ₂ H ₆	C77-H81···N52	1.960	130.758	2.805	C34-H38···N4	1.971	167.94	3.065
	C77-H81···N51	1.998	160.316	3.057	C35-H41···O44	2.137	136.29	3.037
	C77-H79···N28	2.236	149.066	3.230	C34-H36···O15	2.888	148.93	3.883
	C77-H79···N5	2.388	156.829	3.428	C35-H40···N61	3.471	123.23	4.183
	C77-H81···N53	2.395	142.087	3.333	C34-H38···O74	3.938	127.02	4.690
	C78-H84···O65	2.666	144.85	3.622				
	C77-H80···N21	2.751	131.255	3.574				
	C77-H79···N29	2.848	155.325	3.876				
	C77-H80···N60	2.911	141.129	3.831				
	C77-H79···N32	3.329	128.931	4.111				
	C77-H79···N7	3.552	141.687	4.468				
	C78-H83···O16	3.930	142.316	4.848				
CO ₂	C98···O1			3.641	C34···O71			3.231
	C98···O2			3.773	C34···O108			3.883
	C98···N7			3.810	O36···C103			3.337
	C98···O3			3.837	O35···C72			3.800
	C98···N14			3.886	O35···C73			3.805
	O99···C3			3.357	O35···C115			3.808
	O99···C6			3.603	O36···C72			3.864
	O99···C5			3.685				
	O99···C4			3.801				
	O97···C3			3.970				

(4) The chemical stabilities of these materials toward moisture and water are suggested to be examined, which is important to practical industrial applications.

Response: Many thanks for the reviewer's suggestion. To explore the chemical stabilities, we exposed **Zn-fa-datz (1)** and **Zn-fa-atz (2)** to water, moisture (R.H. 35%), and acidic/basic (pH=2/12) aqueous solutions. As shown in **Supplementary Figure 5** and **Figure 15** of the revised version, for **Zn-fa-datz (1)**, after being exposed to water, moisture (*ca.* 35% RH), or acidic aqueous solutions (pH=2) for 7 days, the PXRD and CO₂ uptakes of the treated samples remain almost unchanged, while obvious change was observed in pH=12 after 3 days, demonstrating the exceptional acid stability but poor alkaline stability of **Zn-fa-datz (1)**.

For **Zn-fa-atz (2)**, after being exposed to water, moisture conditions (*ca.* 35% RH) for 7 days, the PXRD peak intensities and N₂ uptakes of the treated samples remain the same, indicating the exceptional chemical stability of **Zn-fa-atz (2)** under aqueous and moisture conditions. However, when **Zn-fa-atz (2)** was exposed to acid (pH=2) or alkaline aqueous (pH=12), the N₂ uptakes of the treated samples dramatically changed after 1 day but the PXRD remain almost unchanged.

The chemical stabilities studies of **Zn-fa-datz (1)** and **Zn-fa-atz (2)** have been added to the main text and presented in the **Supplementary Figure 5** and **Figure 15** of the SI. The modified text was highlighted in red.

Accordingly, the following discussion has been added to the revised main text, also highlighted in red:

“...because of its ultramicroporous nature and polar pore surface without open-metal coordination sites, based on our previously raised general rule.” was revised to “...because of its **high stability in moisture conditions (Supplementary Fig.5)**, ultramicroporous nature and polar pore surface without open-metal coordination sites, based on our previously raised general rule.”

“which is comparable with the value calculated from single-crystal diffraction data at 195 K (0.319 cm³ g⁻¹) (Fig. 3a, Supplementary Fig. 7 and Supplementary Table 1). The pore size distribution analysis based on...” was revised to “which is comparable with the value calculated from single-crystal diffraction data at 195 K (0.319 cm³ g⁻¹) (Fig. 3a, Supplementary Fig. 7 and Supplementary Table 1). **Besides, it is worth to mention that Zn-fa-**

atz (2) can remain the crystalline and porosity after treated with water or exposed to moisture (*ca.* 35%RH) (Supplementary Fig.15). The pore size distribution analysis based on...”

Supplementary Figure 5. PXRD patterns and CO₂ adsorption/desorption isotherms of **Zn-fa-datz (1)** at 195 K after different treatments. After being exposed to water or moisture (*ca.* 35% RH), or acidic aqueous solutions (pH=2) for 7 days, the PXRD and CO₂ uptakes of (Supplementary Fig. 5) the treated samples remain almost unchanged, while obvious change was observed in pH=12 after 3 days, demonstrating the exceptional acid stability but poor alkaline stability of **Zn-fa-datz (1)**.

Supplementary Figure 15. PXRD patterns and N₂ adsorption/desorption isotherms of **Zn-fa-atz (2)** at 77 K after different treatments. After being exposed to water or moisture conditions (*ca.* 35% RH) for 7 days, the PXRD peak intensities and N₂ uptakes of the treated **Zn-fa-atz (2)** samples (Supplementary Fig. 8) remain the same, indicating the exceptional chemical stability of **Zn-fa-atz (2)** under aqueous and moisture conditions. However, when **Zn-fa-atz (2)** was exposed to acid (pH=2) or alkaline aqueous (pH=12), the N₂ uptakes of the treated samples dramatically changed after 1 day but the PXRD remain almost unchanged.

(5) The dynamic breakthrough experiments under humidity conditions are also need to be studied in detail.

Response: Many thanks for the reviewer’s suggestion. As shown in **Supplementary Figure 38**, the dynamic breakthrough experiment was carried out to examine the separation performance of **Zn-fa-atz (2)** in practical scenarios using equimolar CO₂/C₂H₂/C₂H₄/C₂H₆ mixture (1:1:1:1, *v/v/v/v*) under wet (*ca.* 36% RH) conditions. The result showed that **Zn-fa-**

atz (2) can maintain the elution sequence in quaternary $\text{CO}_2/\text{C}_2\text{H}_2/\text{C}_2\text{H}_4/\text{C}_2\text{H}_6$ mixtures (1:1:1:1, v/v/v/v) in the presence of water vapor, but the retention times and the shape of the breakthrough curves have changed significantly, indicating the competitive adsorption of water with the other four gases. Therefore, water would influence the adsorption selectivities of **Zn-fa-atz (2)** for the four gases, because hydrophilic adsorbents (especially those functionalized by open metal sites, including molecular sieves) generally have strong adsorption affinity for these small guest molecules with high polarity.

Note that the competitive adsorption of water is a common problem faced by hydrophilic materials. In many previous reports, the existence of water vapor will seriously affect the separation performance (Figures R4 and R5).

Moreover, we are very thankful for reviewer's suggestion and will investigate the in-depth effect of moisture existed in such sets of dynamic breakthrough experiments in the future work.

Figure R4. Breakthrough curves for an ultramicroporous MOF in dry (left) and 50% RH (right) conditions with N_2/CO_2 (v/v = 50/50) gas mixture. Adopted from *J. Am. Chem. Soc.* **2017**, 139, 1734.

Figure R5. Breakthrough curves for a microporous MOF in a fixed-bed under flow of CO_2/CH_4 (v/v = 50/50, red) and CO_2/N_2 (v/v = 15/85) gas mixtures under dry (black) and wet (blue) (74% RH) environments. Filled circles = CO_2 , empty circles = CH_4 or N_2 . Adopted from *ACS Appl. Mater. Interfaces* **2020**, 12, 41177.

Accordingly, the following discussion has been added to the revised main text:

“However, when **Zn-fa-atz (2)** was exposed to wet quaternary mixtures (ca. 36% RH), the breakthrough curves showed that **Zn-fa-atz (2)** can maintain the elution sequence, but the

retention times and the shape of the breakthrough curves have changed significantly, indicating the competitive adsorption of water with the other four gases (Supplementary Fig. 38).”

Supplementary Figure 38. Experimental breakthrough curves for quaternary $\text{CO}_2/\text{C}_2\text{H}_2/\text{C}_2\text{H}_4/\text{C}_2\text{H}_6$ mixtures (1:1:1:1, v/v/v/v) on **Zn-fa-atz (2)** under wet (*ca.* 36% RH) conditions at room temperature.

Reviewer #2:

Yang et al., present a nice work for the precise pore size control of MOF by regulating the framework hydrogen bond toward complex separation systems. Rational modulation of pore structure is regularly attempted in two- or three-component separation systems; however, this is the first example that is available for the quaternary mixture systems. Obviously, the increase of components will greatly increase the difficulty of separation. This work demonstrates the means of precise control in maneuvering pore sizes at sub-angstrom precision by tailoring the network hydrogen bonds and elucidates the impact of the slight change in pore size on selectivity in a complex system. Impressively, the new MOF can still show a good ability to separate C₂H₄ in one step even in such a complex mixture system, which I thought was very rare in the current research background. Besides, the molded MOF can also maintain good performance, which brings the result closer to industrial application. I believe that the topic and approach described in this manuscript are of general interest to a broad audience. I thus strongly recommend its publication in Nat. Commun., with a few minor points being addressed:

1. In this manuscript, the author calculates the IAST selectivity of the compound and states that the selectivity is not very high. I also believe that in such a complex mixture system, the comparison of IAST selectivity is not dominant, but I still suggest that the authors add IAST data of related compounds in three-component and four-component systems, which will give readers a more intuitive understanding of this separation field.

Response: Many thanks for the reviewer's suggestion. The IAST selectivities of **Zn-fa-datz (1)** and **Zn-fa-atz (2)** were compared with the best-performing sorbents in the C₂H₂/C₂H₄/C₂H₆ three-component system, and C₂H₂/C₂H₄/C₂H₆/CO₂ four-component system (**Supplementary Table 4**). As showed in **Supplementary Table 4**, both **Zn-fa-datz (1)** and **Zn-fa-atz (2)** show moderate selectivity for C₂H₂/C₂H₄, C₂H₆/C₂H₄ and CO₂/C₂H₄. In fact, it is very difficult to maintain the lowest selectivity for C₂H₄ among the four gas components, because the physicochemical properties of the four gases are too similar.

Accordingly, the following discussion has been added to the revised main text:

“The IAST selectivities of **Zn-fa-datz (1)** and **Zn-fa-atz (2)** were compared with the best-performing sorbents in the C₂H₂/C₂H₄/C₂H₆ three-component system, and C₂H₂/C₂H₄/C₂H₆/CO₂ four-component system (Supplementary Table 4). Both **Zn-fa-datz (1)** and **Zn-fa-atz (2)** show moderate selectivity for C₂H₂/C₂H₄, C₂H₆/C₂H₄ and CO₂/C₂H₄. In fact, it is very difficult to maintain the lowest selectivity for C₂H₄ among the four gas components, because the physicochemical properties of the four gases are too similar.”

Supplementary Table 4. Comparison of C₂H₆/C₂H₄, C₂H₂/C₂H₄ and CO₂/C₂H₄ selectivity in porous materials for one-step C₂H₄ purification for three-component, four-component mixtures at 298 K.

Gas mixtures	Adsorbent	IAST selectivity			References
		C ₂ H ₆ /C ₂ H ₄	C ₂ H ₂ /C ₂ H ₄	CO ₂ /C ₂ H ₄	
C ₂ H ₂ /C ₂ H ₄ /C ₂ H ₆	[Zn(BDC)(H ₂ BPZ)]	2.2 ^b	1.6 ^b	-	Angew. Chem. Int. Ed. 2022 , 61 , e202205427
	Zn(ad)(int)	2.4 ^b	1.61 ^a	-	Angew. Chem. Int. Ed. 2022 , 61 , e202208134
	TJT-100	1.2 ^a	1.8 ^a	-	Angew. Chem. Int. Ed. 2018 , 130 , 16299–16303
	Azole-Th-1	1.46 ^b	1.09 ^b	-	Nat. Commun. 2020 , 11 , 3163
	NPU-1	1.32 ^b	1.4 ^b	-	J. Am. Chem. Soc. 2021 , 143 , 1485–1492
	UPC-612	1.4 ^b	1.07 ^b	-	Angew. Chem. Int. Ed. 2021 , 60 , 11350–11358
	UPC-613	1.5 ^b	1.4 ^b	-	
	UIO-67(NH ₂) ₂	1.7 ^b	2.1 ^a	-	J. Am. Chem. Soc. 2022 , 144 , 2614–2623
	Ni-BDC-INA	1.56 ^b	1.37 ^b	-	Chem. Commun. 2022 , 58 , 4954-4957
	CPM-173	1.76 ^b	1.51 ^b	-	
	UiO-66	1.57 ^b	1.45 ^b	-	
	DMOF-1	1.51 ^b	1.35 ^b	-	
	CuTiF ₆ -TPPY	2.12 ^b	5.47 ^b	-	Sci. Adv. 2022 , 8 , eabn9231
	ZJNU-115	1.56 ^b	2.05 ^a	-	Inorg. Chem. 2021 , 60 , 10819-10829
	ZJNU-7	1.56 ^b	1.77 ^a	-	Inorg. Chem. Front. 2021 , 8 , 1243-1252
	NUM-9a	1.62 ^b	1.48 ^a	-	ACS Appl. Mater. Interfaces 2021 , 13 , 962-969
	MIL-125	1.21 ^b	2.32 ^b	-	Sep. Purif. Technol. 2021 , 276 , 119284
	NH ₂ -MIL-125	1.18 ^b	3.75 ^b	-	
	ZSTU-2	1.62 ^b	2.36 ^b	-	
	LIFM-XYX-6	1.63 ^b	1.53 ^a	-	CCS Chem. 2023 , 023.202302698
	UPC-66-a	1.65 ^b	1.05 ^b	-	Chem. 2022 , 8 , 3263–3274
Zn-ATA	1.84 ^a	1.81 ^a	-	Chem. Eng. J. 2022 , 450 , 138272-138277	
HIAM-210	2.0 ^b	2.0 ^b	-	Chem. Sci. 2023 , 14 , 5912–5917	
HIAM-326	1.9 ^b	1.6 ^b	-	Small , 2023 , 2304460	
Zn-fa-datz (1)	1.4 ^b	1.5 ^b	0.8 ^b	This work	
C ₂ H ₂ /C ₂ H ₄ /C ₂ H ₆ /CO ₂	Zn-atz-oba	1.27 ^b	1.43 ^b	1.33 ^b	Nat. Commun. 2021 , 12 , 6507
	Al-MOFM ₁₅ ^c	2.51 ^b	3.32 ^b	-	Chem. Sci. 2022 , 13 , 7172-7180
	Zn-fa-atz (2)	1.6 ^b	1.6 ^b	1.4 ^b	This work

^a IAST selectivity for 1/99 gas mixture. ^b IAST selectivity for 1/1 gas mixture. ^c IAST selectivity calculated at 293K.

2. The hydrogen bond is the reason for the slight pore size difference between Zn-fa-datz (1) and Zn-fa-atz (2), but the related parameters of the framework hydrogen bond in the two structures do not seem to be mentioned in this paper. The manuscript or SI should reflect hydrogen bond parameters and related structural details.

Response: Many thanks for the reviewer's suggestion. The hydrogen bond parameters and related structural details have been added to the revised main text and **Supplementary Figure 12** of the revised SI:

“it can be observed that the pore wall of 1D channel is constituted by fa^{2-} ligands and both two amino groups of datz^- ligands through **four** tight hydrogen-bonding interactions ($\text{O}\cdots\text{H}\cdots\text{N} = 1.95\text{-}2.12 \text{ \AA}$, $\angle\text{O}\cdots\text{H}\cdots\text{N} = 138.8\text{-}170.4^\circ$) (Fig. 2b and Supplementary Fig. 12).”

“For **Zn-fa-atz (2)**, only one side of atz^- ligands are tied to fa^{2-} ligands through two **$\text{O}\cdots\text{H}\cdots\text{N}$** hydrogen bonding interactions ($\text{O}\cdots\text{H}\cdots\text{N} = 2.12\text{-}2.47 \text{ \AA}$, $\angle\text{O}\cdots\text{H}\cdots\text{N} = 135.6\text{-}170.4^\circ$) (Fig. 2d and Supplementary Fig. 12).”

Supplementary Figure 12. The hydrogen bond parameters of **Zn-fa-datz (1)** and **Zn-fa-atz (2)**.

3. The interaction distances between the four guests and the framework are mentioned in Fig. 4. Although hydrogen bonding and electrostatic interactions usually come from multiple directions and are difficult to count for ultramicroporous structures without specific adsorption sites, as a scientific paper, I suggest that clearer interaction distances and angles in a reasonable range (e.g., $< 4 \text{ \AA}$ and $> 120^\circ$ for hydrogen bonds) can be reflected in SI in the form of figures and tables.

Response: Many thanks for the reviewer's suggestion. We list the host-guest interaction details in GCMC simulations and PXRD structural refinements. Detailed interaction angles and

distances are added to the revised SI in **Supplementary Figure 25** and **Tables 7-8** ($< 4 \text{ \AA}$ and $> 120^\circ$ for hydrogen bonds and $< 4 \text{ \AA}$ electrostatic interaction).

Figure 4. GCMC adsorption simulation and adsorbed structures. Preferential $\text{C}_2\text{H}_2/\text{C}_2\text{H}_4/\text{C}_2\text{H}_6/\text{CO}_2$ sorption sites in **Zn-fa-datz (1)** and **Zn-fa-atz (2)** obtained by GCMC simulations. Atom colors: C, gray; H, white; N, blue; O, red; Zn, purple.

Supplementary Figure 25. (Top) the host-guest structures of **Zn-fa-datz (1)** and **Zn-fa-atz (2)** revealed by powder diffraction data from Rietveld refinement analysis; (down) magnified CO_2 -host structures by powder diffraction data from Rietveld refinement analysis.

Supplementary Table 7. Host-guest interactions in GCMC simulated structures of **Zn-fa-datz (1)** and **Zn-fa-atz (2)** loaded with CO₂, C₂H₂, C₂H₄ and C₂H₆.

Guest	Zn-fa-datz (1)				Zn-fa-atz (2)			
		H...A (Å)	D-H...A (°)	D...A (Å)		H...A (Å)	D-H...A (°)	D...A (Å)
C ₂ H ₂	C1-H1...N21	2.942	130.23	3.718	C1-H2...O10	2.534	166.46	3.572
	C1-H1...N22	2.993	174.13	4.051	C2-H1...O4	3.070	144.36	3.980
	C1-H1...N23	3.142	152.70	4.115	C1-H2...N18	3.347	128.88	4.095
	C2-H2...O3	3.161	135.81	3.990				
	C1-H1...N24	3.435	133.92	4.242				
	C2-H2...N14	3.502	141.63	4.382				
C ₂ H ₄	C1-H4...O8	2.677	154.25	3.679	C1-H4...O15	2.581	165.57	3.638
	C2-H3...O1	2.714	172.91	3.788	C2-H3...O5	2.738	164.21	3.790
	C2-H1...O10	2.804	133.48	3.632	C2-H1...O13	2.754	168.91	3.818
	C2-H1...N18	2.840	136.29	3.696	C1-H2...O16	2.912	162.76	3.956
	C2-H1...N15	3.120	141.89	4.024	C2-H1...N28	3.021	123.92	3.732
	C2-H3...N4	3.158	125.66	3.888	C2-H3...O4	3.500	159.82	4.530
	C2-H1...N16	3.285	133.99	4.109	C1-H4...N26	3.578	121.56	4.245
	C1-H2...O13	3.466	155.26	4.471				
C1-H2...N25	3.769	120.97	4.423					
C ₂ H ₆	C1-H4...O4	2.575	168.86	3.649	C1-H2...N5	2.629	141.12	3.542
	C1-H5...O11	2.585	157.18	3.615	C1-H3...O4	2.699	120.23	3.381
	C2-H2...O14	2.793	163.89	3.850	C2-H4...O16	2.708	121.08	3.400
	C2-H3...N35	2.982	146.60	3.937	C1-H3...N1	2.829	151.73	3.823
	C2-H3...N24	3.107	147.66	4.069	C1-H3...N2	3.035	122.02	3.729
	C1-H5...N23	3.229	132.95	4.051	C1-H3...N3	3.175	163.50	4.231
	C1-H5...N22	3.351	124.42	4.068	C1-H2...N4	3.231	135.51	4.079
	C1-H5...O9	3.541	146.71	4.492	C2-H6...N6	3.593	139.37	4.477
	C1-H4...O6	3.723	150.33	4.700	C1-H2...O14	3.753	120.67	4.408
	C2-H1...O12	3.770	134.35	4.597	C2-H6...N19	3.810	163.55	4.865
C2-H2...O7	3.848	151.69	4.833					
CO ₂	C1...O17			3.122	C1...O11			3.023
	C1...N37			3.161	C1...N21			3.407
	C1...N39			3.405	C1...O10			3.679
	C1...N34			3.700	O3...C63			3.299
	C1...O9			3.744	O3...C73			3.354
	O2...C48			3.347	O3...C52			3.597
	O2...C85			3.358	O2...C70			3.892
	O3...C53			3.442	O2...C94			3.979
	O2...C98			3.493				
	O3...C55			3.497				
	O3...C54			3.552				

Supplementary Table 8. Host-guest interactions in PXRD analyses and structural refinements of **Zn-fa-datz (1)** and **Zn-fa-atz (2)** loaded with CO₂, C₂H₂, C₂H₄ and C₂H₆.

Guest molecules	Zn-fa-datz (1)				Zn-fa-atz (2)			
		H...A (Å)	D-H...A (°)	D...A (Å)		H...A (Å)	D-H...A (°)	D...A (Å)
C ₂ H ₂	C51-H53...N48	2.818	163.268	3.856	C24-H26...N94	3.852	121.854	4.499
	C51-H53...N7	3.825	132.105	4.612				
C ₂ H ₄	C50-H53...N44	1.829	164.048	2.895	C93-H96...N125	3.022	126.05	3.747
	C50-H53...N43	2.251	131.006	3.080	C93-H96...O45	3.277	160.87	4.295
	C50-H53...N45	2.564	134.221	3.416	C92-H95...O3	3.507	121.13	4.157
	C50-H52...N21	2.651	163.689	3.712	C92-H94...O46	3.545	149.50	4.516
	C50-H53...N67	2.682	126.992	3.451	C92-H95...N101	3.584	168.60	4.630
	C50-H53...N48	3.148	127.414	3.909	C93-H97...O67	3.987	124.60	4.691
C ₂ H ₆	C77-H81...N52	1.960	130.758	2.805	C34-H38...N4	1.971	167.94	3.065
	C77-H81...N51	1.998	160.316	3.057	C35-H41...O44	2.137	136.29	3.037
	C77-H79...N28	2.236	149.066	3.230	C34-H36...O15	2.888	148.93	3.883
	C77-H79...N5	2.388	156.829	3.428	C35-H40...N61	3.471	123.23	4.183
	C77-H81...N53	2.395	142.087	3.333	C34-H38...O74	3.938	127.02	4.690
	C78-H84...O65	2.666	144.85	3.622				
	C77-H80...N21	2.751	131.255	3.574				
	C77-H79...N29	2.848	155.325	3.876				
	C77-H80...N60	2.911	141.129	3.831				
	C77-H79...N32	3.329	128.931	4.111				
	C77-H79...N7	3.552	141.687	4.468				
	C78-H83...O16	3.930	142.316	4.848				
CO ₂	C98...O1			3.641	C34...O71			3.231
	C98...O2			3.773	C34...O108			3.883
	C98...N7			3.810	O36...C103			3.337
	C98...O3			3.837	O35...C72			3.800
	C98...N14			3.886	O35...C73			3.805
	O99...C3			3.357	O35...C115			3.808
	O99...C6			3.603	O36...C72			3.864
	O99...C5			3.685				
	O99...C4			3.801				
	O97...C3			3.970				

4. The authors use the LDF model to calculate and compare the diffusion coefficients of C₂H₆ in Zn-fa-atz (2) and Zn-fa-atz (2)/PES, but the related calculation methods are not mentioned. Please describe the method.

Response: Many thanks for the reviewer’s suggestion. We are sorry for our unclear description. Accordingly, the following description has been added to **Supplementary Methods 4** of the revised SI, highlighted in red:

“Supplementary Methods 3. Kinetic adsorption measurements and calculation

Time-dependent adsorption determination of CO₂/C₂H₂/C₂H₄/C₂H₆ was performed with a TA Instruments Discovery thermobalance at 298 K and 1 atm or an automatic volumetric adsorption apparatus BELSORP-HP/MAXII at 298 K. Prior to each adsorption measurement, the samples were activated at 75 °C for 0.5/4 h under N₂ or a high vacuum. The resulting curves was used to calculate the diffusional time constants by using the following Eq.1 or Eq.2:

$$\frac{M_t}{M_e} = 1 - \exp(-kt) \quad (\text{Supplementary Equation 1})$$

Where M_t is the mass uptake at time t , M_e is the mass uptake at equilibrium, and k is the kinetic rate constant. [Langmuir 1999, 15, 3206-3218] After normalization of kinetic adsorption data, t (min) and M_t/M_e were taken as x and y axes, respectively.

$$\frac{M_t}{M_e} = \frac{6}{\sqrt{\pi}} \cdot \sqrt{\frac{D}{r^2}} \cdot \sqrt{t} \quad (\text{Supplementary Equation 2})$$

Where M_t is the gas adsorbed amount at time t , M_e is the gas adsorbed amount at equilibrium, D is the diffusivity and r is the radius of the equivalent spherical particle. [J. Am. Chem. Soc. 2011, 133, 5228-5231] Therefore, the slopes of q/q_∞ versus \sqrt{t} are derived from the fitting of the plots in the low gas adsorbed amount range, and then D' is obtained from the square of the slope multiplying by $\pi/36$.”

5. The names of all compounds in the figures should be bold to correspond to the article.

Response: Many thanks for the reviewer’s suggestion. All compound names of the figures have been modified to be bold in the revised main text and SI.

6. Some numerical values are mentioned, but there are no corresponding references.

(1) In the process of ethylene (C₂H₄) production, the product compositions of the thermal decomposition are complicated, and the conversion of dehydrogenation is only ca. 50%–60%.

(2) However, limited by the physicochemical properties of four gas molecules (kinetic

diameter: $\text{CO}_2 \approx \text{C}_2\text{H}_2 < \text{C}_2\text{H}_4 < \text{C}_2\text{H}_6$; quadruple moment: $\text{C}_2\text{H}_2 > \text{CO}_2 > \text{C}_2\text{H}_4 > \text{C}_2\text{H}_6$), it is extremely difficult to separate C_2H_4 from the quaternary $\text{CO}_2/\text{C}_2\text{H}_2/\text{C}_2\text{H}_4/\text{C}_2\text{H}_6$ in one step.

Response: Many thanks for the reviewer's suggestion. The corresponding references have been added into the revised main text:

“(1) In the process of ethylene (C_2H_4) production, the product compositions of the thermal decomposition are complicated, and the conversion of dehydrogenation is only ca. 50%–60%.^[1]”

“(2) However, limited by the physicochemical properties of four gas molecules (kinetic diameter: $\text{CO}_2 \approx \text{C}_2\text{H}_2 < \text{C}_2\text{H}_4 < \text{C}_2\text{H}_6$; quadruple moment: $\text{C}_2\text{H}_2 > \text{CO}_2 > \text{C}_2\text{H}_4 > \text{C}_2\text{H}_6$),^[37-39] it is extremely difficult to separate C_2H_4 from the quaternary $\text{CO}_2/\text{C}_2\text{H}_2/\text{C}_2\text{H}_4/\text{C}_2\text{H}_6$ in one step.”

“

- [1] Yang, Y. *et al.* Ethylene/ethane separation in a stable hydrogen-bonded organic framework through a gating mechanism. *Nature Chem.* **13**, 933-939 (2021).
- [37] Reid, C. R. & Thomas, K. M. Adsorption Kinetics and Size Exclusion Properties of Probe Molecules for the Selective Porosity in a Carbon Molecular Sieve Used for Air Separation. *J. Phys. Chem. B* **105**, 10619-10629 (2001).
- [38] Sircar, S. Basic Research Needs for Design of Adsorptive Gas Separation Processes. *Ind. Eng. Chem. Res.* **45**, 5435-5448 (2006).
- [39] Eguchi, R., Uchida, S. & Mizuno, N. Inverse and High $\text{CO}_2/\text{C}_2\text{H}_2$ Sorption Selectivity in Flexible Organic–Inorganic Ionic Crystals. *Angew. Chem. Int. Ed.* **51**, 1635-1639 (2012).”

Reviewer #3:

This manuscript handles the selective separation of ethylene from multi-component mixtures by tuning the pore control of Zn-fa-atz. As compared with Zn-fa-datz, Zn-fa-atz has relatively larger pores by fewer hydrogen bonds in the host network, as the authors mentioned. Based on single or mixed gas experiments, it is clear that Zn-fa-atz has more affinity with CO₂, and less affinity with ethylene, resulting in more efficient C₂H₄ separation. The results sound great, but this manuscript lacks a discussion. Particularly, there is no discussion of why CO₂ affinity was improved in Zn-fa-atz but C₂H₄ affinity was reduced in detail. The molecular simulations explained them, but molecular simulation results cannot elucidate some possible competitive sorption in the mixed gas. For publication, in my opinion, the discussion section should be strengthened.

Response: Thank you for your kind reminding. We realized that the discussion in our original manuscript is weak and not rigorous enough. Based on the suggestions of the reviewers, we have conducted several experiments and enhanced the discussion on the separation mechanism in the following two aspects: (i) discussion of the competitive adsorption including actual selectivities in breakthrough experiments, kinetic factors, and co-adsorption; (ii) more details in the GCMC and experimental host-guest structures. These results indicate that there are complex competitive adsorption and co-adsorption behaviors in the breakthrough experiment.

(i) Competitive adsorption

We realize that our original description of the pore size is not rigorous enough, so we made corresponding revisions and also the title of the manuscript. In our previous discussion, we claimed that the larger pore size of **Zn-fa-atz (2)** is the reason for the lower C₂H₄ affinity, but the discussion for the enhanced CO₂ affinity is missing or not rigorous.

In fact, for the ultramicroporous structure, it is difficult to describe the interactions between the framework and the guest molecules through a simple visualization concept, because it must have sites oriented from multiple directions. In this work, the differences of Q_{st} between the four gases we studied are already very small, which is also the key challenge in multi-component separation systems. In this case, only a very small difference will change the adsorption order in the breakthrough experiment, and this requires the control of the pore structure to be very accurate.

In ultramicroporous **Zn-fa-datz (1)** and **Zn-fa-atz (2)**, although there is a difference in the number of -NH₂ groups between **Zn-fa-datz (1)** and **Zn-fa-atz (2)**, the -NH₂ groups of both two structures are actually hidden in the frameworks, which makes it difficult to interact with

the guests. Based on the SCXRD analysis, the biggest change caused by amino groups is the number of hydrogen bonds in the framework which further influenced the pore structure (including pore size and pore chemistry) (**Figure 4** and **Supplementary Figure 12**). This influence not only changes the pore sizes but also the spatial arrangement of the adsorption sites in the channel.

In order to clarify the separation mechanism more clearly, we calculated the actual adsorption amount of the adsorbent to each component gas in the breakthrough experiment (**Supplementary Figures 34-35** and **Tables 10-11**). The results show that the actual selectivities are little different from that of IAST selectivities. For **Zn-fa-atz (2)**, the adsorption amount order of each gas is followed as $\text{CO}_2 > \text{C}_2\text{H}_2 > \text{C}_2\text{H}_6 > \text{C}_2\text{H}_4$, being similar with the isotherms, but the selectivity changed in $\text{CO}_2/\text{C}_2\text{H}_4$ ($S_{\text{breakthrough}}=2.17 > S_{\text{IAST}}=1.4$). For **Zn-fa-datx (1)**, the adsorption amount order for each gas is followed as $\text{C}_2\text{H}_2 > \text{C}_2\text{H}_6 > \text{CO}_2 > \text{C}_2\text{H}_4$, which is inconsistent with that of the isotherms ($\text{C}_2\text{H}_2 > \text{C}_2\text{H}_6 > \text{C}_2\text{H}_4 > \text{CO}_2$), and the selectivity of $\text{CO}_2/\text{C}_2\text{H}_4$ also changed ($S_{\text{breakthrough}}=1.27 > S_{\text{IAST}}=0.8$). Obviously, both **Zn-fa-atz (2)** and **Zn-fa-datx (1)** have different degrees of increase in the adsorption of CO_2 in the breakthrough experiments.

We considered that this phenomenon is reasonable, because there are many influencing factors that cannot be reflected by the single-component isotherm in the actual separation scenario, such as the interaction between guest-guest interaction in the framework at the high-pressure section, and the adsorption kinetics of different molecules in mixed systems. Firstly, from a thermodynamic point of view, as shown in **Supplementary Figure 9**, at the low-loading section, the Q_{st} order of adsorption of the four gases by **Zn-fa-atz (1)** is $\text{C}_2\text{H}_6 > \text{C}_2\text{H}_2 > \text{CO}_2 > \text{C}_2\text{H}_4$, while the order change to be $\text{C}_2\text{H}_6 \approx \text{CO}_2 > \text{C}_2\text{H}_2 > \text{C}_2\text{H}_4$ when the loading reaches 2 mmol/g. The Q_{st} of CO_2 does not change obviously with the increase of loading, while the other three gases have a decreasing trend, which is probably caused by the strong guest-guest interaction of CO_2 . This phenomenon has been confirmed by the single crystal structure in the previous study. In the host-guest structure, with the increase of pressure, the C and O atoms of different CO_2 molecules can contact each other through electrostatic attraction, resulting in no significant decrease in the heat of adsorption at high loading (**Figure R2**, ref. *Science*, **2010**, 330, 650). This is only the result inferred from the single-component isotherms, and the adsorption process of the mixture should be more complicated. Moreover, to verify the influence of adsorption kinetics, the diffusion time coefficients (D') of $\text{CO}_2/\text{C}_2\text{H}_2/\text{C}_2\text{H}_4/\text{C}_2\text{H}_6$ in **Zn-fa-atz (1)** were calculated from the adsorption kinetic profiles measured at 298 K. As shown in **Supplementary Figure 36**, the D' value sequence of the four gases is followed as CO_2 (0.3874) \gg C_2H_4 (0.1191) $>$ C_2H_2 (0.0820) $>$ C_2H_6 (0.0478), indicating CO_2 diffused much

faster than other three gases during the breakthrough experiments. Although the Q_{st} of CO₂ is slightly lower than that of C₂H₆, the large diffusion difference causes CO₂ to be adsorbed faster in the breakthrough experiment. Therefore, the larger uptakes of CO₂ are the result of the synergistic effect of adsorption thermodynamics and kinetics.

(ii) More details in the GCMC and experimental host-guest structures

In order to get experimental evidence, PXRD analyses were conducted on gas-loaded **Zn-fa-datz (1)** and **Zn-fa-atz (2)** followed by Rietveld structural refinements to unveil the binding sites of C₂ hydrocarbons and CO₂. Because the single crystals of **Zn-fa-atz (2)** are brittle flakes (**Figure R3**), it was difficult to retain sample single-crystallinity after a series of post-synthetic treatments (exchanging, heating, adsorption, etc.).

We list the host-guest interaction details in GCMC simulations and PXRD structural refinements. Detailed interaction angles and distances are added to the revised SI in **Supplementary Tables 7-8** (< 4 Å and > 120° for hydrogen bonds and < 4 Å electrostatic interaction).

As shown in **Supplementary Figure 25** and **Table 8** of the revised version, all these gas molecules are adsorbed near the pockets enclosed by four triazolate moieties and four fa²⁻ ligands, which is similar with the host-guest structures obtained from GCMC simulations, but there are some differences in structural details. This may be because the GCMC structures present the first adsorption site, while the experimental structures are periodic and obtained at high gas pressure. As shown in **Supplementary Table 8**, for C₂ hydrocarbon guest molecules, it is reasonable that the **Zn-fa-datz (1)** with smaller pore size has a closer interaction with three hydrocarbon molecules. For CO₂ in **Zn-fa-atz (2)**, the optimal position has changed, and the O atom from the framework and the C atom in CO₂ yield a relative strong interaction (C···O = 3.231 Å) that cannot be observed in **Zn-fa-datz (1)**, resulting in higher CO₂ Q_{st} than that of **Zn-fa-datz (1)**.

Accordingly, the following discussion has been added to the revised main text:

The title “Hydrogen Bond Unlocking-Driven Pore Size Control for Shifting Multi-Component Gas Separation Function” was revised to “Hydrogen Bond Unlocking-Driven Pore **Structure** Control for Shifting Multi-Component Gas Separation Function”.

Abstract “Gas adsorption isotherms and molecular simulations indicated that the larger pore size of the newly synthesized coordination network Zn-fa-atz (2) weakened the affinity for three C₂ hydrocarbons synchronously including C₂H₄ but enhanced the affinity to CO₂ with smaller molecular size” was revised to “Gas adsorption isotherms and molecular simulations

indicated that the larger pore size of the newly synthesized coordination network Zn-fa-atz (2) weakened the affinity for three C₂ hydrocarbons synchronously including C₂H₄ but enhanced the affinity to CO₂ **due to the optimized CO₂-host interaction**".

"Hence, we predict that precise pore size control could be achieved by regulating the hydrogen bonds via reducing the amino side groups (i.e., replacing the diamino datz⁻ with unilateral-amino 3-amino-1,2,4-triazolate, atz⁻)" **was revised to** "Hence, we predict that precise pore **structure** control could be achieved by regulating the hydrogen bonds via reducing the amino side groups (i.e., replacing the diamino datz⁻ with unilateral-amino 3-amino-1,2,4-triazolate, atz⁻)"

"Consequently, even a slight change in pore size can significantly affect the affinity, especially for guests with larger molecular sizes.² In case of **Zn-fa-datz (1)** and **Zn-fa-atz (2)**, it can be inferred that the smaller C₂H₂/C₂H₄/C₂H₆ adsorption enthalpy of **Zn-fa-atz (2)** is the outcome of the larger pore size." **was revised to** "Consequently, even a slight change in pore structure can significantly affect the affinity, especially for guests with larger molecular sizes. In case of **Zn-fa-datz (1)** and **Zn-fa-atz (2)**, **although there is a difference in the number of -NH₂ groups between Zn-fa-datz (1) and Zn-fa-atz (2), the -NH₂ groups of both two structures are actually hidden in the frameworks, which makes it difficult to interact with the guests. Based on the SCXRD analysis, the biggest change caused by amino groups is still the number of hydrogen bonds in the framework which further influenced the pore structure (including pore size and pore chemistry) (Figure 4 and Supplementary Fig. 12). This influence not only changes the pore size but also the spatial arrangement of the adsorption sites in the channel.**"

"Because of the very small sizes relative to the aperture, C₂H₂, C₂H₄, and C₂H₆ are all confined in the relatively spacious positions in the cavities of **Zn-fa-datz (1)** and **Zn-fa-atz (2)**, but the molecular orientations change due to the slight difference in pore size." **was revised to** "Because of the very small sizes relative to the aperture, C₂H₂, C₂H₄, and C₂H₆ are all confined in the relatively spacious positions in the cavities of **Zn-fa-datz (1)** and **Zn-fa-atz (2)**, but the molecular orientations change due to the slight difference in pore **structure.**"

"Due to the smaller size than that of the aperture, CO₂ can freely select the optimal positions to fully contact the networks. Therefore, the increase in pore size leads to a dramatic change of the CO₂ molecule location." **was revised to** "Due to the **spatial arrangement of the adsorption sites N/O atoms in the framework, the optimal position of CO₂ was dramatically changed.**"

"To verify the competitive adsorption during the breakthrough experiments, we calculated the actual uptakes of **Zn-fa-datz (1)** and **Zn-fa-atz (2)** for the four gases using the reported method (Supplementary Figs. 34-35 and Tables 10-11) [ref: *Angew. Chem. Int. Ed.* **2019**, *58*, 7692-7696; *Angew. Chem. Int. Ed.* **2020**, *59*, 23322-23328; *Chem* **2021**, *7*, 1006-1019]. The results show that the actual selectivities are little different from that of IAST selectivities. For **Zn-fa-**

atz (2), the adsorption amount order of each gas is followed as $\text{CO}_2 > \text{C}_2\text{H}_2 > \text{C}_2\text{H}_6 > \text{C}_2\text{H}_4$, being similar with the isotherms, but the selectivity changed in $\text{CO}_2/\text{C}_2\text{H}_4$ ($S_{\text{breakthrough}}=2.17 > S_{\text{IAST}}=1.4$). For **Zn-fa-datz (1)**, the adsorption amount order for each gas is followed as $\text{C}_2\text{H}_2 > \text{C}_2\text{H}_6 > \text{CO}_2 > \text{C}_2\text{H}_4$, which is inconsistent with that of the isotherms ($\text{C}_2\text{H}_2 > \text{C}_2\text{H}_6 > \text{C}_2\text{H}_4 > \text{CO}_2$), and the selectivity of $\text{CO}_2/\text{C}_2\text{H}_4$ also changed ($S_{\text{breakthrough}}=1.27 > S_{\text{IAST}}=0.8$). Obviously, both **Zn-fa-atz (2)** and **Zn-fa-datz (1)** have different degrees of increase in the adsorption of CO_2 in the breakthrough experiments. Therefore, the diffusion coefficients of the four gases through the adsorption kinetic profiles at 298 K (Supplementary Fig. 36) were calculated. The results showed a that the diffusion of CO_2 (0.3874) was significantly higher than that of C_2H_4 (0.1191), C_2H_2 (0.0820), and C_2H_6 (0.0478), indicating CO_2 diffused much faster than other three gases during the breakthrough experiments. Therefore, the larger uptakes of CO_2 are the result of the synergistic effect of adsorption thermodynamics and kinetics.”

“Additionally, the gas-loaded structures were also studied by the PXRD analyses and corresponding refinements (Supplementary Fig. 25-29 and Table 8). The results are similar with the host-guest structures obtained from GCMC simulations, but there are some differences in structural details. This may because the simulated structures present the first adsorption site, while the experimental structures are periodic and obtained at high gas pressure. For C_2 hydrocarbon guest molecules, it is reasonable that the **Zn-fa-atz (2)** with smaller pore size has a closer interaction with the guests. For CO_2 in **Zn-fa-atz (2)**, the optimal position has changed, and the O atom from the framework can contact closely with the C atom in CO_2 , yielding a relative strong interaction ($\text{C}\cdots\text{O} = 3.231 \text{ \AA}$) that cannot be observed in **Zn-fa-datz (1)**, resulting in higher CO_2 Q_{st} than that of **Zn-fa-datz (1)**.”

“In conclusion, fine-tuning of pore size by regulating the network hydrogen bonding interactions in two related coordination networks can precisely control the adsorption selectivity of C_2H_4 in the complex separation systems.” was revised to “In conclusion, fine-tuning of pore **structure** by regulating the network hydrogen bonding interactions in two related coordination networks can precisely control the adsorption selectivity of C_2H_4 in the complex separation systems.”

Details of PXRD analyses on gas-loaded structures were added to supplementary “Pawley and Rietveld refinement of PXRD” section of the SI, and the host-guest structures and related refinement parameters have been listed in **Supplementary Figures 25-29** of the SI, all of which highlighted in red.

Figure 4. GCMC adsorption simulation and adsorbed structures. Preferential $C_2H_2/C_2H_4/C_2H_6/CO_2$ sorption sites in **Zn-fa-datz (1)** and **Zn-fa-atz (2)** obtained by GCMC simulations. Atom colors: C, gray; H, white; N, blue; O, red; Zn, purple.

Supplementary Figure 12. The hydrogen bond parameters of **Zn-fa-datz (1)** and **Zn-fa-atz (2)**.

Supplementary Figure 34. Experimental breakthrough curves for He of 2 mL/min **Zn-fa-atz (2)** at room temperature and 1 bar.

Supplementary Figure 35. Experimental breakthrough curves of **Zn-fa-datz (1)** and **Zn-fa-atz (2)** for quaternary $\text{CO}_2/\text{C}_2\text{H}_2/\text{C}_2\text{H}_4/\text{C}_2\text{H}_6$ mixtures (1:1:1:1, v/v/v/v) with velocity correction.

Supplementary Figure 9. Gas adsorption enthalpies of **Zn-fa-datz (1)** (a), **Zn-fa-atz (2)** (b) calculated by virial method.

Figure R2. Direct observation of the guest-guest interaction between two different CO_2 molecules in a microporous MOF by single-crystal X-ray analysis. Adopted from ref. *Science*, **2010**, 330, 650.

Supplementary Figure 36. Adsorption kinetics profiles (point) and linear fittings (line) of CO₂ (black), C₂H₂ (red), C₂H₄ (blue) and C₂H₆ (green) for **Zn-fa-atz (2)** at 298 K.

Figure R3. Optical microscope image of **Zn-fa-atz (2)** crystal.

Supplementary Figure 25. (Top) the host-guest structures of **Zn-fa-datz (1)** and **Zn-fa-atz (2)** revealed by powder diffraction data from Rietveld refinement analysis; (down) magnified CO₂-host structures by powder diffraction data from Rietveld refinement analysis.

Supplementary Figure 26. Rietveld refinement plots of powder X-ray diffraction data of CO₂-loaded **Zn-fa-datz (1)** and C₂H₂-loaded **Zn-fa-datz (1)**.

Supplementary Figure 27. Rietveld refinement plots of powder X-ray diffraction data of C₂H₄-loaded **Zn-fa-datz (1)** and C₂H₆-loaded **Zn-fa-datz (1)**.

Supplementary Figure 28. Rietveld refinement plots of powder X-ray diffraction data of CO₂-loaded **Zn-fa-atz (2)** and C₂H₂-loaded **Zn-fa-atz (2)**.

Supplementary Figure 29. Rietveld refinement plots of powder X-ray diffraction data of C₂H₄-loaded **Zn-fa-atz (2)** and C₂H₆-loaded **Zn-fa-atz (2)**.

Supplementary Table 7. Host-guest interactions in GCMC simulated structures of **Zn-fa-datz (1)** and **Zn-fa-atz (2)** loaded with CO₂, C₂H₂, C₂H₄ and C₂H₆.

Guest	Zn-fa-datz (1)				Zn-fa-atz (2)			
		H···A (Å)	D-H···A (°)	D···A (Å)		H···A (Å)	D-H···A (°)	D···A (Å)
C ₂ H ₂	C1-H1···N21	2.942	130.23	3.718	C1-H2···O10	2.534	166.46	3.572
	C1-H1···N22	2.993	174.13	4.051	C2-H1···O4	3.070	144.36	3.980
	C1-H1···N23	3.142	152.70	4.115	C1-H2···N18	3.347	128.88	4.095
	C2-H2···O3	3.161	135.81	3.990				
	C1-H1···N24	3.435	133.92	4.242				
	C2-H2···N14	3.502	141.63	4.382				
C ₂ H ₄	C1-H4···O8	2.677	154.25	3.679	C1-H4···O15	2.581	165.57	3.638
	C2-H3···O1	2.714	172.91	3.788	C2-H3···O5	2.738	164.21	3.790
	C2-H1···O10	2.804	133.48	3.632	C2-H1···O13	2.754	168.91	3.818
	C2-H1···N18	2.840	136.29	3.696	C1-H2···O16	2.912	162.76	3.956
	C2-H1···N15	3.120	141.89	4.024	C2-H1···N28	3.021	123.92	3.732
	C2-H3···N4	3.158	125.66	3.888	C2-H3···O4	3.500	159.82	4.530
	C2-H1···N16	3.285	133.99	4.109	C1-H4···N26	3.578	121.56	4.245
	C1-H2···O13	3.466	155.26	4.471				
C1-H2···N25	3.769	120.97	4.423					
C ₂ H ₆	C1-H4···O4	2.575	168.86	3.649	C1-H2···N5	2.629	141.12	3.542
	C1-H5···O11	2.585	157.18	3.615	C1-H3···O4	2.699	120.23	3.381
	C2-H2···O14	2.793	163.89	3.850	C2-H4···O16	2.708	121.08	3.400
	C2-H3···N35	2.982	146.60	3.937	C1-H3···N1	2.829	151.73	3.823
	C2-H3···N24	3.107	147.66	4.069	C1-H3···N2	3.035	122.02	3.729
	C1-H5···N23	3.229	132.95	4.051	C1-H3···N3	3.175	163.50	4.231
	C1-H5···N22	3.351	124.42	4.068	C1-H2···N4	3.231	135.51	4.079
	C1-H5···O9	3.541	146.71	4.492	C2-H6···N6	3.593	139.37	4.477
	C1-H4···O6	3.723	150.33	4.700	C1-H2···O14	3.753	120.67	4.408
	C2-H1···O12	3.770	134.35	4.597	C2-H6···N19	3.810	163.55	4.865
	C2-H2···O7	3.848	151.69	4.833				
CO ₂	C1···O17			3.122	C1···O11			3.023
	C1···N37			3.161	C1···N21			3.407
	C1···N39			3.405	C1···O10			3.679
	C1···N34			3.700	O3···C63			3.299
	C1···O9			3.744	O3···C73			3.354
	O2···C48			3.347	O3···C52			3.597
	O2···C85			3.358	O2···C70			3.892
	O3···C53			3.442	O2···C94			3.979
	O2···C98			3.493				
	O3···C55			3.497				
	O3···C54			3.552				

Supplementary Table 8. Host-guest interactions in PXRD analyses and structural refinements of **Zn-fa-datz (1)** and **Zn-fa-atz (2)** loaded with CO₂, C₂H₂, C₂H₄ and C₂H₆.

Guest molecules	Zn-fa-datz (1)			Zn-fa-atz (2)				
		H...A (Å)	D-H...A (°)	D...A (Å)		H...A (Å)	D-H...A (°)	D...A (Å)
C ₂ H ₂	C51-H53...N48	2.818	163.268	3.856	C24-H26...N94	3.852	121.854	4.499
	C51-H53...N7	3.825	132.105	4.612				
C ₂ H ₄	C50-H53...N44	1.829	164.048	2.895	C93-H96...N125	3.022	126.05	3.747
	C50-H53...N43	2.251	131.006	3.080	C93-H96...O45	3.277	160.87	4.295
	C50-H53...N45	2.564	134.221	3.416	C92-H95...O3	3.507	121.13	4.157
	C50-H52...N21	2.651	163.689	3.712	C92-H94...O46	3.545	149.50	4.516
	C50-H53...N67	2.682	126.992	3.451	C92-H95...N101	3.584	168.60	4.630
	C50-H53...N48	3.148	127.414	3.909	C93-H97...O67	3.987	124.60	4.691
C ₂ H ₆	C77-H81...N52	1.960	130.758	2.805	C34-H38...N4	1.971	167.94	3.065
	C77-H81...N51	1.998	160.316	3.057	C35-H41...O44	2.137	136.29	3.037
	C77-H79...N28	2.236	149.066	3.230	C34-H36...O15	2.888	148.93	3.883
	C77-H79...N5	2.388	156.829	3.428	C35-H40...N61	3.471	123.23	4.183
	C77-H81...N53	2.395	142.087	3.333	C34-H38...O74	3.938	127.02	4.690
	C78-H84...O65	2.666	144.85	3.622				
	C77-H80...N21	2.751	131.255	3.574				
	C77-H79...N29	2.848	155.325	3.876				
	C77-H80...N60	2.911	141.129	3.831				
	C77-H79...N32	3.329	128.931	4.111				
	C77-H79...N7	3.552	141.687	4.468				
	C78-H83...O16	3.930	142.316	4.848				
CO ₂	C98...O1			3.641	C34...O71			3.231
	C98...O2			3.773	C34...O108			3.883
	C98...N7			3.810	O36...C103			3.337
	C98...O3			3.837	O35...C72			3.800
	C98...N14			3.886	O35...C73			3.805
	O99...C3			3.357	O35...C115			3.808
	O99...C6			3.603	O36...C72			3.864
	O99...C5			3.685				
	O99...C4			3.801				
	O97...C3			3.970				

Supplementary Table 10. The CO₂/C₂H₂/C₂H₄/C₂H₆ breakthrough performances of **Zn-fa-datz (1)**.

Zn-fa-datz (1)	CO ₂ uptake (mmol/g)	C ₂ H ₂ uptake (mmol/g)	C ₂ H ₄ uptake (mmol/g)	C ₂ H ₆ uptake (mmol/g)	S (C ₂ H ₆ /C ₂ H ₄)	S (C ₂ H ₂ /C ₂ H ₄)	S (CO ₂ /C ₂ H ₄)
Adsorption isotherm at 0.25 bar (298 K)	1.54	2.31	1.69	1.97	-	-	-
Breakthrough experiment	0.52	0.73	0.41	0.63	1.6	1.6	0.8
IAST	-	-	-	-	1.54	1.78	1.27

[a] The single-component gas adsorption uptake data calculated by dual-site Langmuir-Freundlich fittings at partial pressure of 0.25 bar for C₂ gases.

Supplementary Table 11. The CO₂/C₂H₂/C₂H₄/C₂H₆ breakthrough performances of **Zn-fa-atz (2)**.

Zn-fa-atz (2)	CO ₂ uptake (mmol/g)	C ₂ H ₂ uptake (mmol/g)	C ₂ H ₄ uptake (mmol/g)	C ₂ H ₆ uptake (mmol/g)	S (C ₂ H ₆ /C ₂ H ₄)	S (C ₂ H ₂ /C ₂ H ₄)	S (CO ₂ /C ₂ H ₄)
Adsorption isotherm at 0.25 bar (298 K)	1.5	1.5	1.3	1.4	-	-	-
Breakthrough experiment at 1 bar (298 K)	0.50	0.46	0.23	0.43	1.87	2.00	2.17
IAST	-	-	-	-	1.4	1.5	1.4

Supplementary Methods 9. Pawley and Rietveld refinement of PXRD

The microcrystalline **Zn-fa-datz (1)** and **Zn-fa-atz (2)** was placed in a glass capillary ($\Phi = 0.8$ mm) connected with an automatic volumetric sorption apparatus (Micromeritics 3FLEX), and heated under high vacuum at 75 °C for 4 hours. After that, CO₂, C₂H₂, C₂H₄ and C₂H₆ gas was introduced by cooling the samples with dry ice-acetone bath to 195 K, and the gas dosed volumetrically from calibrated pressure. PXRD data of the gas-loaded samples was collected on a Rigaku SmartLab X-ray powder diffractometer (Cu K α) with a scanning speed of 0.01 °/step and 7 s/step under capillary transmission mode. All the indexing and refinement were performed by the Reflex plus module of Material Studio 5.0. The pseudo-Voigt profile parameters, background parameters, the cell parameters, the zero point of the diffraction pattern, the global isotropic atom displacement parameters, the Berar-Baldinozzi asymmetry correction parameters, and the March-Dollase preferred orientation correction parameters were optimized step by step to improve the agreement between the calculated and the experimental powder diffraction patterns.

REVIEWER COMMENTS

Reviewer #1 (Remarks to the Author):

The revised manuscript by Yang et al. has addressed most of the reviewers' comments, and they conducted PXRD analyses on gas-loaded Zn-fa-datz and Zn-fa-atz to unveil the gas binding sites. The authors still claimed that the slight change in pore size can significantly affect the affinity of guest molecules, and thus lead to the reversed CO₂ adsorption. However, Only 0.2 Å size difference (6.1 and 6.3 Å for Zn-fa-datz and Zn-fa-atz) can really induce so large CO₂ and C₂ adsorption difference? From the gas-loaded PXRD data (Figure S25), it is more likely that the reduction of amino groups in Zn-fa-atz can induce obvious changes on the local pore shape and chemistry, which makes the CO₂ and C₂ molecule change their binding position and binding model obviously, rather than the pore size control. So their core claim on "enlarging the pore size at sub-angstrom precision for shifting multi-component gas separation" may be not so accurate. In addition, the adsorption kinetics was also proved to play the important role on the selective adsorption of CO₂. The authors cannot simply ascribe the mechanism to pore size control by hydrogen bond unlocking, such as in Fig. 1. And some of the current claims on the separation mechanism might be not accurate. Given the separation performance have no progress than the literatures and the strategy is not so general, whether this work has enough impact and novelty to be published or not can be decided by the editors.

Other suggestions:

The figures in this manuscript are still low quality. For example, the crystal structure depictions in Fig. 2 are rough and breezing. It is very difficult for the readers to understand what's the structural and coordination differences between Zn-fa-datz and Zn-fa-atz, and the hydrogen bonding unlocking means in this figure.

The authors are suggested to present CO₂ and C₂ binding sites by the experimental PXRD analyses into the main text, especially for CO₂ adsorption site.

Reviewer #2 (Remarks to the Author):

The authors have revised the paper thoroughly according to three reviewers' comments. I am satisfied with the revised version, which can be accepted.

Reviewer #1:

The revised manuscript by Yang et al. has addressed most of the reviewers' comments, and they conducted PXRD analyses on gas-loaded Zn-fa-datz and Zn-fa-atz to unveil the gas binding sites. The authors still claimed that the slight change in pore size can significantly affect the affinity of guest molecules, and thus lead to the reversed CO₂ adsorption. However, Only 0.2 Å size difference (6.1 and 6.3 Å for Zn-fa-datz and Zn-fa-atz) can really induce so large CO₂ and C₂ adsorption difference? From the gas-loaded PXRD data (Figure S25), it is more likely that the reduction of amino groups in Zn-fa-atz can induce obvious changes on the local pore shape and chemistry, which makes the CO₂ and C₂ molecule change their binding position and binding model obviously, rather than the pore size control. So their core claim on “enlarging the pore size at sub-angstrom precision for shifting multi-component gas separation” may be not so accurate. In addition, the adsorption kinetics was also proved to play the important role on the selective adsorption of CO₂. The authors cannot simply ascribe the mechanism to pore size control by hydrogen bond unlocking, such as in Fig. 1. And some of the current claims on the separation mechanism might be not accurate. Given the separation performance have no progress than the literatures and the strategy is not so general, whether this work has enough impact and novelty to be published or not can be decided by the editors.

Response: Thank you very much for the valuable comments of the reviewers. Based on these suggestions, the quality of the manuscript has been significantly improved. We realize that the previous discussion on the mechanism of different separation properties of the two structures is not accurate. This point has been corrected in the revised manuscript and discussed in detail based on the experimental data. The revisions were shown as follows:

“Gas adsorption isotherms and molecular simulations indicated that the larger pore size of the newly synthesized coordination network **Zn-fa-atz (2)** weakened the affinity for three C₂ hydrocarbons synchronously including C₂H₄ but enhanced the affinity to CO₂

due to the optimized CO₂-host interaction, leading to effective C₂H₄ production from the CO₂/C₂H₂/C₂H₄/C₂H₆ mixture in one step based on the experimental and simulated breakthrough data.” was revised to “Gas adsorption isotherms, adsorption kinetics and gas-loaded crystal structure analysis indicated that the newly synthesized coordination network **Zn-fa-atz (2)** weakened the affinity for three C₂ hydrocarbons synchronously including C₂H₄ but enhanced the CO₂ adsorption due to the optimized CO₂-host interaction and the faster CO₂ diffusion, leading to effective C₂H₄ production from the CO₂/C₂H₂/C₂H₄/C₂H₆ mixture in one step based on the experimental and simulated breakthrough data.”

“Single-crystal X-ray diffraction revealed that reducing the amino groups on an aminotriazolate-based ligand resulted in less hydrogen bonds in the host network, enlarging the pore size at sub-angstrom precision.” was revised to “Single-crystal X-ray diffraction revealed that reducing the amino groups on an aminotriazolate-based ligand resulted in less hydrogen bonds in the host network, which led to changes in the pore shape and pore chemistry.”

“Herein, we show that unlocking the framework hydrogen bonding can affect the pore structure, and weaken the affinity to C₂ hydrocarbons, especially C₂H₄ (Fig. 1).” was revised to “Herein, we show that unlocking the framework hydrogen bonding can affect the pore shape and pore chemistry, and weaken the affinity to C₂ hydrocarbons, especially C₂H₄ (Fig. 1).”

“We speculated that by fine tuning the pore size, it is possible to reverse the adsorption affinity of C₂H₄ and CO₂ without affecting the adsorption sequence of C₂H₂/C₂H₄/C₂H₆.” was revised to “We speculated that by fine tuning the pore structure to achieve a more optimized CO₂ adsorption environment, it is possible to reverse the adsorption affinity of C₂H₄ and CO₂ without affecting the adsorption sequence of C₂H₂/C₂H₄/C₂H₆.”

“The hydrogen bonds restrict the swing of ligands and narrow the cavities. Hence, we

predict that precise pore structure control could be achieved by regulating the hydrogen bonds via reducing the amino side groups (i.e., replacing the diamino datz⁻ with unilateral-amino 3-amino-1,2,4-triazolate, atz⁻)." was revised to "The hydrogen bonds restrict the swing of ligands and **determine the arrangement of adsorption sites and shape of the channel**. Hence, we predict that precise pore structure control could be achieved by regulating the hydrogen bonds via reducing the amino side groups (i.e., replacing the diamino datz⁻ with unilateral-amino 3-amino-1,2,4-triazolate, atz⁻)."

"For **Zn-fa-atz (2)**, only one side of atz⁻ ligands are tied to fa²⁻ ligands through two O-H...N hydrogen bonding interactions (O-H...N = 2.12-2.47 Å, ∠O-H...N = 135.6-170.4°) (Fig. 2d and Supplementary Fig. 12). but in **Zn-fa-datz (1)**, both sides of datz⁻ ligand can connect with fa²⁻ ligands by four hydrogen bonds. Debonding the hydrogen bonds causes the rotation of the five-member ring of atz⁻ ligand, resulting in different dihedral angles between atz⁻/datz⁻ and Zn-atz/datz layers (Supplementary Fig. 2). For clarity, the diagonals between four adjacent O atoms from different fa²⁻ ligands were used to compare the pore sizes of the two structures (minus the van der Waals radius of O atom of 1.52 Å) (Figs. 2a and 2c).⁵⁶ As expected, the 1D channel of **Zn-fa-atz (2)** (5.5 × 4.9 Å) is slightly larger than that of **Zn-fa-datz (1)** (5.4 × 4.6 Å)." was revised to "For **Zn-fa-atz (2)**, only one side of atz⁻ ligands are tied to fa²⁻ ligands through two O-H...N hydrogen bonding interactions (O-H...N = 2.12-2.47 Å, ∠O-H...N = 135.6-170.4°) (Fig. 2h and Supplementary Fig. 12). but in **Zn-fa-datz (1)**, both sides of datz⁻ ligand can connect with fa²⁻ ligands by four hydrogen bonds. **When the diagonals between four adjacent O atoms from different fa²⁻ ligands were used to compare the pore sizes of the two structures (minus the van der Waals radius of O atom of 1.52 Å) (Supplementary Fig. 2)⁵⁵, it can be seen that the difference in aperture between **Zn-fa-atz (2)** (5.5 × 4.9 Å) and **Zn-fa-datz (1)** (5.4 × 4.6 Å) is very small. In fact, debonding the hydrogen bonds causes the rotation of the five-member ring of atz⁻ ligand, resulting in different dihedral angles between atz⁻/datz⁻ and Zn-atz/datz layers (Figs. 2c and 2g). Therefore, the greater difference between the **Zn-fa-atz (2)** and **Zn-fa-datz (1)** is reflected in the shape of the pore and the local pore chemistry."**

“The pore size distribution analysis based on the Horvath-Kawazoe model also reveals that the 1D channels of **Zn-fa-atz (2)** (6.3 Å) is slightly larger than that of **Zn-fa-datx (1)** (6.1 Å), which is consistent with single-crystal analysis.” was revised to “The pore size distribution analysis based on the Horvath-Kawazoe model also reveals that the 1D channels of **Zn-fa-atz (2)** (6.3 Å) comparable with that of **Zn-fa-datx (1)** (6.1 Å), which is consistent with single-crystal analysis.”

“Consequently, even a slight change in pore size can significantly affect the affinity, especially for guests with larger molecular sizes. In case of **Zn-fa-datx (1)** and **Zn-fa-atz (2)**, although there is a difference in the number of -NH₂ groups between **Zn-fa-datx (1)** and **Zn-fa-atz (2)**, the -NH₂ groups of both two structures are actually hidden in the frameworks, which makes it difficult to interact with the guests. Based on the SCXRD analysis, the biggest change caused by amino groups is still the number of hydrogen bonds in the framework which further influenced the pore structure (including pore size and pore chemistry) (Figure 4 and Supplementary Fig. 12). This influence not only changes the pore size but also the spatial arrangement of the adsorption sites in the channel.” was revised to “Consequently, even a slight change in the local pore shape and chemistry can significantly affect the affinity. In case of **Zn-fa-datx (1)** and **Zn-fa-atz (2)**, based on the SCXRD analysis, the decrease of the amino groups not only changes the local chemical environment of the pore, but also reduce hydrogen bonds in the framework which further leads to the change in the shape of the channel (the spatial arrangement of the adsorption sites). (Fig. 2 and Supplementary Fig. 2 and 12).”

“Additionally, the gas-loaded structures were also studied by the PXRD analyses and corresponding refinements (Supplementary Fig. 25-29 and Table 8). The results are similar with the host-guest structures obtained from GCMC simulations, but there are some differences in structural details. This may be because the simulated structures present the first adsorption site, while the experimental structures are periodic and obtained at

high gas pressure. For C2 hydrocarbon guest molecules, it is reasonable that the **Zn-fa-datz (1)** with smaller pore size has a closer interaction with three C2 hydrocarbon molecules. For CO₂ in **Zn-fa-atz (2)**, the optimal position has changed, and the O atom from the framework can contact closely with the C atom in CO₂, yielding a relative strong interaction (C···O = 3.231 Å) that cannot be observed in **Zn-fa-datz (1)**, resulting in higher CO₂ Q_{st} than that of **Zn-fa-datz (1)**.” was revised to “To further understand the role of pore structure tuning, the host–guest structures of **Zn-fa-datz (1)** and **Zn-fa-atz (2)** were studied by the PXRD analyses and corresponding refinements (Fig. 4, Supplementary 17-20 and Table 5). The eight studied host–guest systems showed that all the gas molecules preferentially localized within the pockets enclosed by four triazolate moieties and four fa^{2-} ligands. For C₂H₂/C₂H₄/C₂H₆, the host–guest interactions are mainly contributed by weak O/N···H–C hydrogen bonding interactions from multiple orientations. C₂H₂, C₂H₄, and C₂H₆ are all confined in the relatively spacious positions in the cavities of **Zn-fa-datz (1)** and **Zn-fa-atz (2)**, but the molecular orientations change due to the difference in pore shape and pore chemistry. As shown in Fig. 4 and Table 5, most measured O/N···H–C distances in **Zn-fa-atz (2)** are slightly longer than that in **Zn-fa-datz (1)**, which is consistent with the synchronous decrease of the Q_{st} for the three C2 hydrocarbons in **Zn-fa-atz (2)**. For CO₂ in **Zn-fa-atz (2)** (Fig. 4), the optimal position has changed when compared with that in **Zn-fa-datz (1)**, and the O atom from the framework can contact closely with the C atom in CO₂, yielding a relative strong interaction (C···O = 3.231 Å) that cannot be observed in **Zn-fa-datz (1)**, resulting in higher CO₂ Q_{st} than that of **Zn-fa-datz (1)**.”

“In conclusion, fine-tuning of pore structure by regulating the network hydrogen bonding interactions in two related coordination networks can precisely control the adsorption selectivity of C₂H₄ in the complex separation systems.” was revised to “In conclusion, fine-tuning of **local pore shape and chemistry** by regulating the network hydrogen bonding interactions in two related coordination networks can precisely control the adsorption selectivity of C₂H₄ in the complex separation systems.”

Other suggestions:

(1) The figures in this manuscript are still low quality. For example, the crystal structure depictions in Fig. 2 are rough and breezing. It is very difficult for the readers to understand what's the structural and coordination differences between Zn-fa-datz and Zn-fa-atz, and the hydrogen bonding unlocking means in this figure.

Response: Thank you very much for the advice of the reviewers. We have made corresponding changes in **Fig. 2**. The comparison of the Zn-azole layers with two pillared-layer structures are added to show the effect of different number of amino groups (**Figs. 2b** and **2f**). **Fig. 2c** and **Fig. 2g** show the difference of local pore shape and chemical environment between **Zn-fa-datz (1)** and **Zn-fa-atz (2)**. Fig. 2d and **Fig. 2h** show the reason for the difference between **Zn-fa-datz (1)** and **Zn-fa-atz (2)** channels, that is, the effect of hydrogen bonds in the framework. In addition, **Fig. 1** also removed the original discussion about pore size and added a description of pore shape and chemistry.

Fig. 1 Illustration of hydrogen bond unlocking-driven pore shape and chemistry control to shift multi-component separation.

Fig. 2 Crystal structures of **Zn-fa-datz (1)** (a-d) and **Zn-fa-atz (2)** (e-h). Perspective view of the structure along the 1D channels of **Zn-fa-datz (1)** (a) and **Zn-fa-atz (2)** (e). Zinc-aminotriazolate layer of **Zn-fa-datz (1)** (b) and **Zn-fa-atz (2)** (f). Dihedral angles in **Zn-fa-datz (1)** (c) and **Zn-fa-atz (2)** (g) between $atz^-/datz^-$ and Zn- $atz/datz$ layers. Front views of pore walls with highlighted (yellow) H-N \cdots O interactions of **Zn-fa-**

datz (1) (d) and **Zn-fa-atz (2) (h)**. Color code: Zn, purple; C, gray; O, red; N, blue; H, white.

(2) The authors are suggested to present CO₂ and C₂ binding sites by the experimental PXRD analyses into the main text, especially for CO₂ adsorption site.

Response: Many thanks for the reviewer's suggestion. Accordingly, we replace the host-guest structures from GCMC simulation with that from experimental PXRD analyses in **Fig. 4**, and the corresponding discussions were also revised:

Fig. 4 The host-guest structures of **Zn-fa-datz (1)** and **Zn-fa-atz (2)** revealed by powder diffraction data from Rietveld refinement analysis. Color code: Zn, purple; C, gray; O, red; N, blue; H, white.

“Additionally, the gas-loaded structures were also studied by the PXRD analyses and corresponding refinements (Supplementary Fig. 25-29 and Table 8). The results are similar with the host-guest structures obtained from GCMC simulations, but there are some differences in structural details. This may be because the simulated structures present the first adsorption site, while the experimental structures are periodic and obtained at high gas pressure. For C₂ hydrocarbon guest molecules, it is reasonable that the **Zn-fa-datz (1)** with smaller pore size has a closer interaction with three C₂ hydrocarbon

molecules. For CO₂ in **Zn-fa-atz (2)**, the optimal position has changed, and the O atom from the framework can contact closely with the C atom in CO₂, yielding a relative strong interaction (C···O = 3.231 Å) that cannot be observed in **Zn-fa-datz (1)**, resulting in higher CO₂ Q_{st} than that of **Zn-fa-datz (1)**.” was revised to “To further understand the role of pore structure tuning, the host–guest structures of **Zn-fa-datz (1)** and **Zn-fa-atz (2)** were studied by the PXRD analyses and corresponding refinements (Fig. 4, Supplementary 17-20 and Table 5). The eight studied hos–guest systems showed that all the gas molecules preferentially localized within the pockets enclosed by four triazolate moieties and four fa²⁻ ligands. For C₂H₂/C₂H₄/C₂H₆, the host–guest interactions are mainly contributed by weak O/N···H–C hydrogen bonding interactions from multiple orientations. C₂H₂, C₂H₄, and C₂H₆ are all confined in the relatively spacious positions in the cavities of **Zn-fa-datz (1)** and **Zn-fa-atz (2)**, but the molecular orientations change due to the difference in pore shape and pore chemistry. As shown in Fig. 4 and Table 5, most measured O/N···H–C distances in **Zn-fa-atz (2)** are slightly longer than that in **Zn-fa-datz (1)**, which is consistent with the synchronous decrease of the Q_{st} for the three C2 hydrocarbons in **Zn-fa-atz (2)**. For CO₂ in **Zn-fa-atz (2)** (Fig. 4), the optimal position has changed when compared with that in **Zn-fa-datz (1)**, and the O atom from the framework can contact closely with the C atom in CO₂, yielding a relative strong interaction (C···O = 3.231 Å) that cannot be observed in **Zn-fa-datz (1)**, resulting in higher CO₂ Q_{st} than that of **Zn-fa-datz (1)**.”

REVIEWERS' COMMENTS

Reviewer #2 (Remarks to the Author):

I have reviewed the former versions of this paper several months ago. After careful evaluation, I recommend the acceptance of this revised paper. In fact, I cannot agree the comments from Reviewer #1 that "From the gas-loaded PXRD data (Figure S25), it is more likely that the reduction of amino groups in Zn-fa-atz can induce obvious changes on the local pore shape and chemistry, which makes the CO₂ and C₂ molecule change their binding position and binding model obviously, rather than the pore size control." Under this synthesis conditions, the reduction of amino groups is not easy. Also, even the amino groups could be reduced, the reduction process will also be occurred for the Zn-fa-datz material with two amino groups. So, it is unreasonable that the reduction of amino groups as the key factors for the adsorption difference on CO₂ and C₂ molecules. In my opinion, as claimed in the revised paper, different amino groups will lead to changes in the pore shape and pore chemistry except for the pore size, which are important for the adsorption difference on CO₂ and C₂ molecules.

Reviewer #2:

I have reviewed the former versions of this paper several months ago. After careful evaluation, I recommend the acceptance of this revised paper. In fact, I cannot agree the comments from Reviewer #1 that “From the gas-loaded PXRD data (Figure S25), it is more likely that the reduction of amino groups in Zn-fa-atz can induce obvious changes on the local pore shape and chemistry, which makes the CO₂ and C₂ molecule change their binding position and binding model obviously, rather than the pore size control.” Under this synthesis conditions, the reduction of amino groups is not easy. Also, even the amino groups could be reduced, the reduction process will also be occurred for the Zn-fa-datz material with two amino groups. So, it is unreasonable that the reduction of amino groups as the key factors for the adsorption difference on CO₂ and C₂ molecules. In my opinion, as claimed in the revised paper, different amino groups will lead to changes in the pore shape and pore chemistry except for the pore size, which are important for the adsorption difference on CO₂ and C₂ molecules.

Response: Thank you for your comments. We agree with the reviewers' consideration of the change of amino groups on the pore size/shape and pore chemistry in the two structures. According to the comments of the reviewers, and to further avoid ambiguity, we have corrected some of the statements in the manuscript:

“Single-crystal X-ray diffraction revealed that reducing the amino groups on an aminotriazolate-based ligand resulted in less hydrogen bonds in the host network, which led to changes in the pore shape and pore chemistry.” was revised as “Single-crystal X-ray diffraction revealed that the different amino groups on the triazolate ligands resulted in the change of the hydrogen bonding in the host network, which led to changes in the pore size/shape and pore chemistry.”

“Herein, we show that unlocking the framework hydrogen bonding can affect the pore shape and pore chemistry...” was revised as “Herein, we show that unlocking the framework hydrogen bonding can affect the pore size/shape and pore chemistry...”

“The hydrogen bonds restrict the swing of ligands and determine the arrangement of adsorption sites and shape of the channel. Hence, we predict that precise pore structure control could be achieved by regulating the hydrogen bonds via reducing the amino side groups (i.e., replacing the diamino datz^- with unilateral-amino 3-amino-1,2,4-triazolate, atz^-).” was revised as “The hydrogen bonds restrict the swing of ligands and determine the arrangement of adsorption sites and **size/shape** of the channel. Hence, we predict that precise pore structure control could be achieved by regulating the hydrogen bonds via **different amino side groups** (i.e., replacing the diamino datz^- with unilateral-amino 3-amino-1,2,4-triazolate, atz^-).”

“Therefore, the greater difference between the **Zn-fa-atz (2)** and **Zn-fa-datz (1)** is reflected in the shape of the pore and the local pore chemistry.” was revised as “Therefore, the greater difference between the **Zn-fa-atz (2)** and **Zn-fa-datz (1)** is reflected in the **size/shape of the pore** and the local pore chemistry.”

“Consequently, even a slight change in the local pore shape and chemistry can significantly affect the affinity. In case of **Zn-fa-datz (1)** and **Zn-fa-atz (2)**, based on the SCXRD analysis, the decrease of the amino groups not only changes the local chemical environment of the pore, but also reduce hydrogen bonds in the framework which further leads to the change in the shape of the channel (the spatial arrangement of the adsorption sites).” was revised as “Consequently, even a slight change in the **pore size/shape and local pore chemistry** can significantly affect the affinity. In case of **Zn-fa-datz (1)** and **Zn-fa-atz (2)**, based on the SCXRD analysis, **different amino groups** not only change the local chemical environment of the pore, but also **affect hydrogen bonds** in the framework which further leads to the change in the **size/shape of the channel** (the spatial arrangement of the adsorption sites).”

“In conclusion, fine-tuning of local pore shape and chemistry by regulating the network hydrogen bonding interactions in two related coordination networks can...” was revised

as “In conclusion, fine-tuning pore size/shape and local pore chemistry by regulating the network hydrogen bonding interactions in two related coordination networks can...”

Figure 1 was updated as follows:

Fig. 1 Illustration of hydrogen bond unlocking-driven pore size/shape and chemistry control to shift multi-component separation (Color code: metal nodes, white; guest molecule, orange; H-bonding single-site/dual-site donors, blue; H-bonding acceptors, red; H bonding, black dotted line; weak interaction, orange dotted line; the direction of gas flow, orange row; derivation of structure-function relationship, black row).